# Koina: Democratizing machine learning for proteomics research

Ludwig Lautenbacher [1,2,20], Kevin L. Yang [3,20], Tobias Kockmann [4], Christian Panse [4,5], Wassim Gabriel [1], Dulguun Bold[1], Elias Kahl[1], Matthew Chambers[6], Brendan X. MacLean [6], Kai Li [3], Fengchao Yu [7], Brian C. Searle [8,9,10], Damien Beau Wilburn[10,11], Mohammad Reza Zare Shahneh [12], Yuhui Hong [13], Haixu Tang [13], Mingxun Wang [12,14], Ralf Gabriels [15,16], Robbin Bouwmeester [15,16], Robbe Devreese [15,16], Jesse Angelis [1], Eduard Sabidó [17,18], Tobias K. Schmidt [19], Alexey I. Nesvizhskii [3,7] ✉ & Mathias Wilhelm [1,2] ✉

Recent developments in machine learning (ML) and deep learning have immense potential for applications in proteomics, such as generating spectral libraries, improving peptide identification, and optimizing targeted acquisition modes. Although new ML models are regularly published, the rate at which the community adopts these models is slow. This is in part due to a lack of findability and accessibility of these models as well as the technical challenges involved in incorporating these models into data analysis pipelines and demonstrating their reusability for end-users. Here we show Koina, an open-source decentralized and online-accessible model repository to facilitate publication of ML models. Koina enables ML model usage via an easy-to-use online interface, facilitating the integration of ML models in data analysis pipelines. Using the widely used FragPipe computational platform as an example, we demonstrate how Koina can be integrated with existing proteomics software tools and how these integrations improve data analysis.

Recent advancements in machine learning (ML) and deep learning (DL) hold substantial promise for numerous applications in proteomics, particularly in tasks such as generating optimized spectral libraries[1] and enhancing peptide identification in data-dependent acquisition (DDA)[2–6], data-independent acquisition (DIA)[7], and targeted acquisition modes[8]. Despite the frequent publication of new ML/DL models (hereafter abbreviated to ML models) designed specifically for proteomics, their adoption

[1]Computational Mass Spectrometry, Technical University of Munich (TUM), Freising, Germany. [2]Munich Data Science Institute, Technical University of Munich, Garching, Germany. [3]Gilbert S. Omenn Department of Computational Medicine and Bioinformatics, University of Michigan, Ann Arbor, MI, USA. [4]Functional Genomics Center Zurich (FGCZ) - University of Zurich | ETH Zurich, Winterthurerstrasse 190, CH-, Zurich, Switzerland. [5]Swiss Institute of Bioinformatics (SIB), Quartier Sorge - Batiment Amphipole, CH-, Lausanne, Switzerland. [6]Department of Genome Sciences, University of Washington, Seattle, WA, USA. [7]Department of Pathology, University of Michigan, Ann Arbor, MI, USA. [8]Pelotonia Institute for Immuno-Oncology, The Ohio State University Comprehensive Cancer Center, Columbus, Ohio, USA. [9]Department of Biomedical Informatics, The Ohio State University, Columbus, Ohio, USA. [10]Department of Chemistry and Biochemistry, The Ohio State University, Columbus, Ohio, USA. [11]Center for RNA Biology, The Ohio State University, Columbus, OH, US. [12]Department of Computer Science, University of California Riverside, Riverside, CA, USA. [13]Luddy School of Informatics, Computing, and Engineering, Indiana University Bloomington, Bloomington, IN, USA. [14]Virtual Multi-Omics Laboratory, The Internet, Riverside, CA, USA. [15]VIB-UGent Center for Medical Biotechnology, VIB, Ghent, Belgium. [16]Department of Biomolecular Medicine, Ghent University, Ghent, Belgium. [17]Centre for Genomic Regulation (CRG), The Barcelona Institute of Science and Technology (BIST), Dr. Aiguader 88, Barcelona, Spain. [18]Universitat Pompeu Fabra (UPF), Dr. Aiguader 88, Barcelona, Spain. [19]MSAID GmbH, Garching, Germany. [20]These authors contributed equally: Ludwig Lautenbacher, Kevin L. Yang. ✉e-mail: nesvi@med.umich.edu; mathias.wilhelm@tum.de

within the community remains confined to a limited number of popular model ecosystems.

To better understand the factors influencing this limited adoption, it is helpful to consider the ML lifecycle, which encompasses three key stages: data preparation, model training, and model deployment. Notably, the model deployment stage is often undervalued in scientific contexts despite its crucial role in translating models into tangible benefits for the community. Addressing this is essential not only for users who seek to integrate state-of-the-art models into their data analysis pipelines to enhance their outcomes but also for ML developers who encounter significant challenges when attempting to benchmark their newly developed models against existing standards. We believe that the models most widely utilized by researchers have achieved success largely because the authors invested a substantial amount of time and effort into making models available via well-curated codebases[6] or dedicated web servers[9,10]. However, these efforts are model-specific and don't allow integration of other models, requiring the same investment for each new model. Furthermore, it creates an increasing number of APIs that third-party tools must maintain if they wish to use state-of-the-art ML models.

These challenges highlight the need for a standardized approach to model deployment to streamline access and usability. One potential solution is establishing a model repository that houses pre-trained machine learning models, readily available for download and application[11–13]. However, while this approach enhances availability, it does not entirely address other interoperability challenges end users face. For example, since Python is the dominant language for developing ML models, users of other popular programming languages such as R, Java, JavaScript and C# encounter significant barriers when attempting to access models, even if they are available in model repositories. Furthermore, hardware limitations associated with deep learning create disparities within the scientific community, putting those without the resources to maintain the necessary infrastructure at a disadvantage. To enhance the effective use of ML models, the FAIR guidelines provide a valuable framework aimed at improving the findability, accessibility, interoperability, and reusability of these resources. Adopting these guidelines to direct the publication of ML models will foster collaboration and promote resource sharing, ultimately benefiting the broader scientific community.

In this work, we introduce Koina—a web-accessible model repository designed to enhance access to machine-learning models for predicting peptide properties used in proteomics. Importantly, Koina exceeds the traditional model repository approach found in other fields by not only allowing models to be used locally but also providing web-based access. This dual approach merges the accessibility of a model repository with the ease of use of a web service. We believe this strategy holds significant potential to democratize machine learning, enabling laboratories with limited resources and expertise to leverage high-performance computing infrastructure for their data analysis needs. Moreover, by improving accessibility to these state-of-the-art ML models, we improve collaboration across different tools available to the scientific community by allowing both ML and downstream tool developers to focus on their individual expertise while still benefiting from each other's work. Furthermore, we demonstrate how Koina simplifies the benchmarking of various ML models when used alongside MSFragger and MSBooster[4], enhancing peptide-spectrum-match (PSM) rescoring within the FragPipe computational proteomics platform. In addition to this benchmarking study, we introduce an algorithm designed to identify optimal models for PSM rescoring, facilitating optimal peptide and protein identification results for end users.

## Results

### Koina: a catalyst for accessible machine learning

Koina is a model repository designed to democratize access to ML models specifically for bottom-up proteomics analysis. By enabling the remote execution of ML models, Koina generates predictions in response to HTTPS requests, which are the standard protocols used for nearly all web traffic. This design choice enhances accessibility, allowing users to use ML models in any programming language without requiring specialized software and hardware. Such an approach not only facilitates the efficient sharing of centralized hardware resources but also permits easy horizontal scaling to accommodate varying demands from the user base (Fig. 1a). Furthermore, we maintain a public network of Koina instances at (https://koina.wilhelmlab.org) to minimize barriers to entry, distributing computational workloads across processing nodes hosted by various research institutions and spin-offs throughout Europe. The intention is to ensure just-in-time results delivery, thereby aligning with the principles of open and collaborative science. We envision scaling this public Koina-Network to meet the evolving needs of the research community, with research groups or institutions contributing additional hardware resources as needed, democratizing access to state-of-the-art ML models in the process. This decentralized architecture raises data security questions, particularly for sensitive clinical datasets. To address this, we provide a Docker image (Fig. 1a) that facilitates the deployment of a self-hosted instance within a secured network, ensuring no external communication. This option allows Koina to be utilized while supporting stringent data security measures.

The initial release of Koina specifically targets the analytical challenges of bottom-up proteomics. It offers access to models that predict a) fragmentation behavior in the gas phase under various conditions (MS/MS), b) chromatographic separation behavior in the liquid phase (RT), c) ion mobility of peptides, and d) peptide detectability. Specifically, Koina supports more than 30 models from the Prosit[9,14–16], MS2PIP[3,17–19], and AlphaPeptDeep[6] (PeptDeep) ecosystems, along with standalone models like DeepLC[20], IM2Deep[18], UniSpec[21], and Chronologer[22] (Fig. 1a). We selected these models not only for their widespread use within the community but also for their development by independent research groups and because they represent a diverse array of modeling approaches and training data choices, thus enabling better analysis across different types of experimental data.

A critical function of any model repository is to provide users with an overview of the state-of-the-art models available. Koina excels in this regard by offering concise, user-friendly documentation accessible online at (https://koina.wilhelmlab.org/docs), significantly simplifying the discovery of new proteomics ML models as well as enabling users to quickly judge which model is most suitable for their use case. Notably, this documentation is semi-automatically generated from an annotation file using OpenAPI and DOME standards[23,24], ensuring that ML developers can contribute new models without extensive web development knowledge. We aim to foster a collaborative community where machine learning developers can contribute their newly developed models to Koina. By doing so, we hope to increase accessibility and maximize the impact of these models, ultimately benefiting everyone working in proteomics.

One of the primary challenges users face when leveraging ML models is a lack of interoperability[25]. Different developers often utilize unique input and output formats, complicating the process for users who must adapt their inputs accordingly. In the field of proteomics, this is particularly cumbersome for peptide sequences, given the multitude of available notation systems. Additionally, variations in supported post-translational modifications (PTMs) and peptide lengths across different models exacerbate the problem. To address these challenges, we developed a simple query interface (Fig. 1b) that standardizes interactions between all available models, based on the ProForma 2.0 format developed by the Proteomics Standards Initiative (PSI) in collaboration with members of the Consortium for Top-Down Proteomics (CTDP)[26,27]. Ideally, users should not need to change their client-side code to use a new model. To achieve this, we first

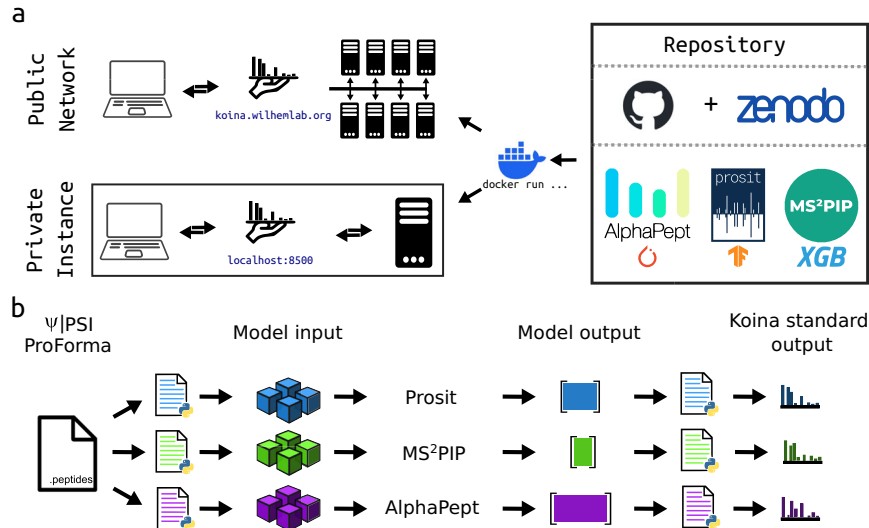

**Fig. 1 | Architecture overview of Koina. a** Overview of the Koina web-accessible model repository. All code used for Koina is publicly available on GitHub. Model weights are stored on Zenodo and fetched dynamically on server startup. A Docker image is made available via the GitHub container registry. It allows for easy scaling & deployment of private instances. The web service design of Koina allows requests from any source, such as KoinaPy (Python), KoinaR (Rlang), Oktoberfest (Python), FragPipe, EncyclopeDIA, Scaffold DIA (Java), Skyline (C#), or any other programming language, simplifying access from these languages for all currently implemented machine-learning models. Koina supports models from all major ML development frameworks. Currently, Koina offers most models from the Prosit, MS2PiP and PeptDeep ecosystems, as well as several standalone models. **b** Schematic of a peptide sequence interface for all models available via Koina to standardize pre- and post-processing steps based on a common input format, namely the PSI ProForma 2.0 peptide notation standard, improving model interoperability. PyTorch, the PyTorch logo and any related marks are trademarks of The Linux Foundation. TensorFlow, the TensorFlow logo and any related marks are trademarks of Google Inc.

standardize the data types and formats for both inputs and outputs. Next, we focus on standardizing input formats; while this is generally easy for most data types, peptide sequences require special attention. We have created new pre-processing steps that convert the standard ProForma 2.0 sequence format into the specific formats needed by different models. For outputs, we aim to keep the original format to allow users to check their predictions against their original sources. We expect this approach will have a minimal effect on interoperability, as most metrics used in later data analysis are not significantly impacted by differences in spectrum normalization or retention times. This shared interface encapsulates the technically heterogeneous collections of models and their associated pre- and post-processing steps in independent computational units, abstracting unnecessary detail for end users in the form of a "workflow" or "execution graph" (Fig. 2). Pre- and post-processing steps can be written in Python, enabling developers to re-use most, if not all, of their code when publishing a model on Koina. Notably, the pre- and post-processing steps are designed as independent computational units, allowing for the re-use between models (Fig. 2) as well as the parallel execution of models, improving performance and inference latency. Consequently, Koina's interface substantially improves interoperability between models, allowing rapid integration of all models available on Koina in third-party applications.

The encapsulated, directly executable models guarantee that dependencies are explicitly encoded, thereby establishing a solid foundation for the long-term reusability of ML models. We support this with a continuous integration (CI) pipeline using GitHub actions, allowing changes to pre- and post-processing scripts—such as performance optimizations or dependency updates—without introducing unintended effects on prediction reproducibility. Reproducibility is further supported via GitHub's transparent change tracking and the release of separate Docker images for every version of the model. Version control is also facilitated through Zenodo, which stores the model files that are not tracked via git, ensuring that files cannot be altered without creating a new version. Additionally, Koina accommodates the simultaneous hosting of different versions of models, allowing users to access both the latest versions and previously released iterations. These design choices foster better reproducibility of published models while maintaining flexibility for publishing bug fixes or performance improvements. The importance of reproducibility in research and model development cannot be overstated. The importance of reproducibility in research and model development cannot be overstated[28,29]. Reproducibility ensures that findings can be verified and trusted[29], which is crucial in scientific work where decisions may rely on model predictions[25]. It fosters transparency and accountability, enabling other researchers to build upon existing work with confidence. A clear commitment to reproducibility provides a safeguard against errors and discrepancies that may arise during implementation or deployment[29]. Hence, our emphasis on reproducibility not only strengthens our own work but also contributes to the larger scientific community by fostering shared knowledge and collaborative advancement.

**Leveraging Koina to integrate ML in proteomics software**
The language-agnostic nature of Koina allows it to be accessed using multiple different methods depending on the user's specific needs, enabling many more people to utilize ML to assist with their data analysis than what would be possible without it. Koina simplifies the client's role by managing all of the model-related logic on the server, leaving the client responsible only for minimal tasks related to sending HTTP/S requests, such as input formatting, batching, request preparation, and error handling. To further improve usability, we developed client packages for two of the most widely used data science languages, Python (KoinaPy available via PyPI) and R (KoinaR available via Bioconductor). These client packages allow users to generate predictions with as few as four lines of code (Supplementary Fig. 1a) in a readily usable format (Supplementary Fig. 1b). Additionally, we provide example code snippets at (https://koina.wilhelmlab.org), demonstrating how to obtain predictions for these client packages and other popular languages like Java, C# and JavaScript.

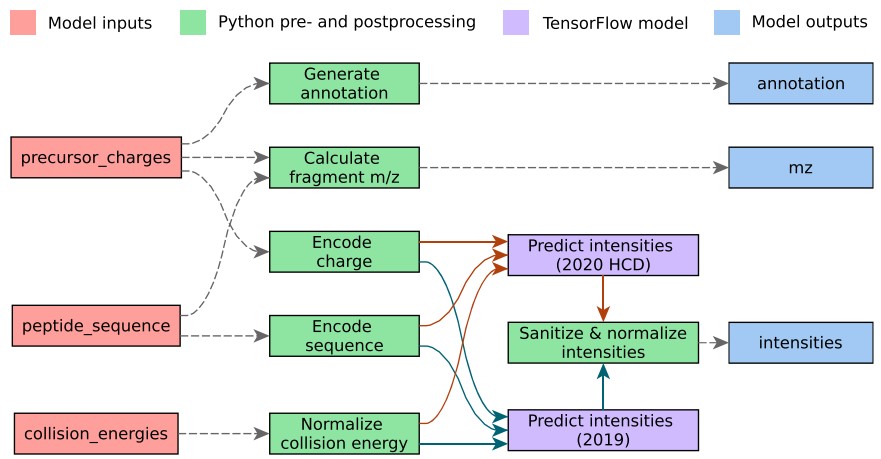

**Fig. 2 | Execution graphs for the Prosit_2019_intensity and the Prosit_2020_intensity_HCD models.** Inputs are colored red, pre- and post-processing steps are colored green and outputs are blue. The TensorFlow neural network is purple. Arrows indicate where data is transferred between models. The gray dashed lines indicate identical steps between the two execution graphs. The solid-colored lines show steps unique for each model.

Koina has already been integrated with several popular proteomics data analysis software, such as Skyline[30], EncyclopeDIA[31], Oktoberfest[2] and FragPipe[4,32,33] (Fig. 3a). These integrations enable users to apply deep learning in application-oriented downstream tasks like spectral library generation or PSM rescoring. Notably, due to the consistent interface between models, integration with Koina allows seamless interoperability between all models currently available on Koina and models released in the future as well.

Oktoberfest is an open-source Python package developed for generating spectral libraries, conducting collision energy (CE) calibration, and executing PSM rescoring. Initially, these features were part of a web service associated with Prosit[9]. In contrast to the original web service, Oktoberfest presents notable advantages: it is open-source, accommodates multiple search engines, and allows for the parallel rescoring of multiple searches. Koina has largely enabled the transition of this formerly closed-source workflow into an open-source Python package. This transition also broadens Oktoberfest's capabilities by supporting predictions from various machine learning models, rather than being restricted to only Prosit models, thus enhancing its versatility and performance. For users aiming to integrate machine learning into their proteomics workflows, Oktoberfest offers both an application programming interface (API) and a command line interface (CLI), promoting seamless integration. However, some familiarity with Python is required. Available on GitHub and installable via PyPI, Oktoberfest ensures straightforward access for researchers and developers.

Koina is also integrated into EncyclopeDIA as a generic predicted library generation tool to support a wide variety of DDA and DIA library search engines. Through the application, users supply a FASTA database as well as parameters such as digestion enzyme constraints and desired charge states. Each peptide/charge state pair that passes the specified constraints is sent to Koina to predict retention times, ion mobility, and fragmentation patterns in separate network packets. The prediction interface is batched, highly threaded, and robustly engineered to account for network errors common in large proteome-sized prediction tasks. EncyclopeDIA also provides a wide variety of library conversion options that support a plethora of external tools, including Spectronaut, DIA-NN[7], Scribe[34], SpectraST[35], MSPepSearch[36], and Skyline/BiblioSpec[37]. Unique to DIA workflows, collision energies can be automatically adjusted based on peptide charge state to account for all peptides being co-fragmented, assuming a single normalized collision energy. Lastly, Koina is supported at both the command line and inside the EncyclopeDIA graphical user interface (GUI) to simplify both automated and manual workflows.

In contrast, the integration of Koina into Skyline enables users to predict MS/MS fragment ion intensities and peptide retention time (RT) for targeted peptides. This alleviates the need for time-consuming and potentially expensive DDA experiments to acquire spectral libraries. Predictions are made on the fly as needed to enable a transparent user experience. As a result, users can develop targeted assays without needing DDA runs and possibly even synthetic peptides for verification. While at the moment, only a subset of models available on Koina are supported and there is no possibility to set an alternative Koina server as a source for predictions, this integration nevertheless saves users time and money and makes Skyline a more powerful tool for proteomics research. Skyline also provides a seamless wrapper for EncyclopeDIA for direct DIA analysis, which uses Koina to predict FASTA-sized libraries as needed. Both Skyline and EncyclopeDIA are open-source tools under the Apache 2 license, making them convenient templates for efficiently implementing Koina using C# and Java.

FragPipe is a comprehensive computational platform for analyzing proteomics data. It is powered by a fast MSFragger[32] database search engine capable of both conventional peptide identification searches and "open" post-translational modification analyses. FragPipe includes the MSBooster module for DL-based rescoring of PSMs coupled with Percolator[38] and Philosopher[33] for protein inference and false discovery rate (FDR) filtering. FragPipe provides complete data analysis workflows for processing quantitative proteomics data generated using all major DDA and DIA-based experimental strategies. We selected FragPipe to provide a detailed demonstration of how an existing computational pipeline, and more specifically, the open-source MSBooster module of FragPipe, can be integrated with Koina.

Even prior to this work, FragPipe included an ML-based rescoring tool called MSBooster[4]. However, the integration with Koina now enables MSBooster to access many more models for prediction besides the previous default, the prediction module of DIA-NN[7], allowing users to compare and, more importantly, mix and match models all within the same software environment. The FragPipe GUI Validation Tab has been extended to include a Koina section (Supplementary Fig. 2a). MSBooster takes as input the list of PSMs reported by MSFragger (in.pin format). It filters out peptides too long for prediction, and if there are peptides with PTMs unsupported by a model, the unmodified version is queried, and its fragment m/z values are adjusted as previously described[4]. MSBooster can query and parse predictions from Koina for thousands of peptides per second using the default public server at (https://koina.wilhelmlab.org) (Fig. 3c, and Supplementary Data 1). When timing on a high-performance Linux server, the DIA-NN prediction module runs faster than the Koina

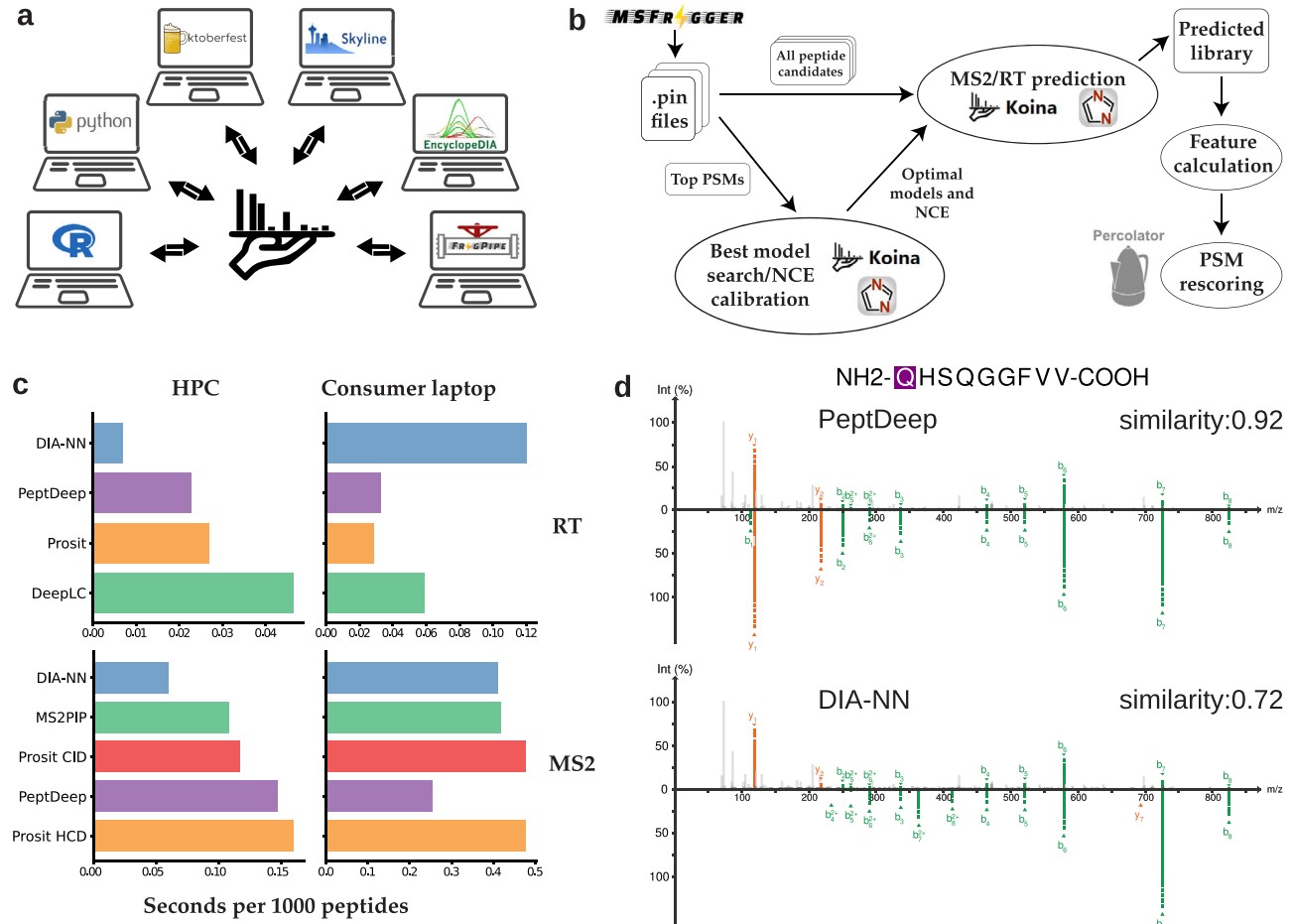

**Fig. 3 | Koina is easily integrated into third-party software. a** Overview of the most notable paths to fetch predictions from Koina. **b** MSBooster workflow of FragPipe for PSM rescoring. All peptide candidates are extracted from .pin files and predicted by either DIA-NN or models available on Koina. These models can be specified manually and NCE calibration is performed automatically if the model accepts NCE as a parameter. MSBooster can also use a heuristic algorithm to automatically choose the best-performing MS/MS + RT model combination. The one thousand top PSMs are selected by default for the best model search and NCE calibration steps. **c** Timing performed using MSBooster. Figures are separated by type of model (RT or MS/MS) and computing environment used. **d** Spectra 20200317_QE_HFX2_LC3_DDA_CM647_R01. 42711.42711.2 from Pak et al., 2021 was visualized using either PeptDeep or DIA-NN predicted spectra, with the experimental spectra on the top and predicted spectra on the bottom. Unweighted spectral entropy was used as the MS/MS similarity metric. Source data are provided as a Source Data file. R logo released under a [Creative Commons Attribution-ShareAlike 4.0 International License (CC BY-SA 4.0)].

models. When timing the models on a consumer laptop, DIA-NN predicts RT slower than the Koina models and PeptDeep even predicts MS/MS spectra faster than DIA-NN, suggesting that DIA-NN's performance is more dependent on the local hardware compared to Koina's performance. These results are consistent with an in-depth throughput and latency benchmark we performed. Depending on the used model, available Koina instances, and client-side concurrency, a throughput of more than 200,000 predictions per second (median > 30,000) was achieved (Supplementary Fig. 3–6), with only a subset of the available computational resources available to the full Koina-Network. The scaling behavior we observed also shows that with the addition of additional hardware, higher performance can be achieved for most models (Supplementary Fig. 7–14). To account for differences between the learned normalized collision energy (NCE) patterns of the predictors and the NCE settings on individual instruments, MSBooster performs an NCE calibration step when using Koina models (Fig. 3b, Methods). To assess the quality of the rescoring, MSBooster generates multiple plots, including a plot showing the rate of Koina prediction (Supplementary Fig. 2b) and a plot showing the distributions of the similarity scores for the PSMs at each NCE value (Supplementary Fig. 2c).

A predicted library is saved as a Mascot Generic Format (MGF) file if a Koina model was used or as a binary file if DIA-NN's prediction module was used. The libraries can be loaded into FragPipe-PDV[39] viewer of FragPipe to compare experimental and predicted spectra (Fig. 3d). Here, it becomes easier to visualize the differences between models' predictions. For example, DIA-NN does not predict intensities for fragments shorter than three amino acids long, as they are less informative for DIA-NN's peptide-centric approach. Figure 3d shows one example in which the pyroglutamated peptide [U:Gln->pyro-Glu]-QHSQGGFVV exhibits a strong y1 fragment intensity, which is matched by PeptDeep's prediction but not by DIA-NN's, resulting in a nearly 0.2 drop in unweighted spectral entropy, which ranges from 0 to 1.

## PSM Rescoring using Koina and MSBooster

Koina in MSBooster enables users to compare model performance systematically by keeping all other upstream (MSFragger) and downstream (Percolator rescoring using MSBooster generated .pin files, protein inference and FDR filtering with Philosopher) data processing steps and parameters the same. Because of differences in model architectures and training data, we hypothesize that the optimal model for rescoring will depend on the specifics of the dataset. To investigate

this, we considered various experiments to pinpoint patterns between specific proteomics data types and optimal models. Specifically, we considered phosphoproteomics data from Arabidopsis thaliana[40] and mouse pancreatic ductal adenocarcinoma cell lines[41], DDA and DIA human leukocyte antigen (HLA) data across different instruments[42–45], DIA data generated on the Orbitrap Astral mass spectrometer (Thermo Scientific)[46,47], and TMT 11 plex-labeled data[16,48] (Supplementary Data 2). We considered the spectral similarity and RT difference features calculated by MSBooster individually and together for PSM rescoring and found that generally, PeptDeep performs best for phosphoproteomics data (Supplementary Fig. 15 & 16), Prosit for HLA data, and DIA-NN for Astral data. Prosit and DIA-NN's prediction module performed comparably on the TMT datasets (Supplementary Fig. 17 & 18). We summarize our results on the HLA data here as an example use case where PSM rescoring particularly benefited from Koina's models.

It is well established that HLA immunopeptidomics data benefits greatly from PSM rescoring with predicted libraries[3,6,14,15,49–51], in part

because the larger nonspecific peptide search space leads to many more ambiguous identifications[52]. HLA peptides fall into one of two classes based on which class of major histocompatibility complex (MHC) molecule they bind, similarly designated class I or II. They have been widely studied because of increased interest in HLA peptides for use as immunotherapeutics[53,54]. Recent developments in DIA and instrumentation (such as the development of the Bruker timsTOF and Thermo Scientific Orbitrap Astral instruments) have provided deeper and more reproducible views of the HLA peptidome[17,55]. We benchmarked all models on a variety of HLA datasets (Fig. 4a-g, Supplementary Data 2). Overall, a combination of the DIA-NN's prediction module and Prosit RT models performed the best over baseline, while the Prosit models excelled in the MS/MS category. The largest gain enabled by MSBooster occurred in a DIA dataset[42], where Prosit rescoring added 53.2% more peptide identifications. In addition, we found that combining the best individual RT and MS/MS models outperformed the current FragPipe default of using DIA-NN's predictions to compute both the MS/MS and RT features for all datasets except for one of the DIA datasets (Fig. 4e).

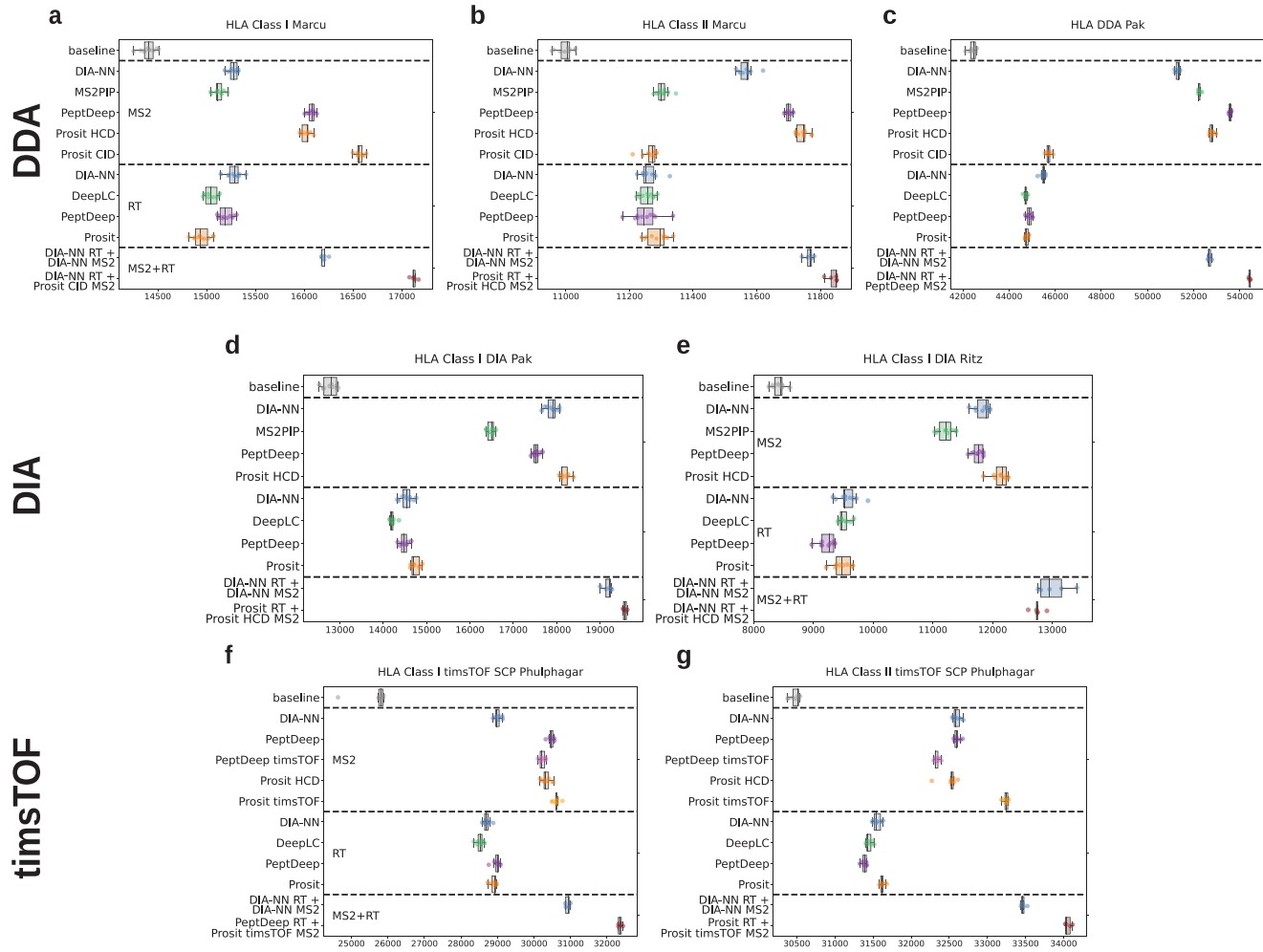

**Fig. 4 | The performance of PSM rescoring of multiple HLA datasets depends on the model used.** The datasets used are (**a**, **b**) Marcu et al., class I and II[43], **c**, **d** Pak et al., class I DDA and DIA[42], **e** Ritz et al., class I DIA[44], and **f**, **g** Phulphagar et al. class I and II on a timsTOF single-cell proteomics system[45]. "PeptDeep timsTOF" is the PeptDeep model with "timsTOF" set as its instrument metadata, while "PeptDeep" used either "Lumos" or "QE" as its instrument metadata, depending on what was listed in the mzML files or manuscript text of the respective datasets. "Baseline" signifies the peptides identified when excluding MSBooster before Percolator rescoring. MSBooster calculated and added the unweighted spectral entropy feature (MS2), delta RT loess feature (RT), or both (MS2 + RT). Boxplots show the interquartile range (IQR) and median, with whiskers at 1.5 times the IQR below the first quartile and above the third quartile. Swarmplots are plotted on top, and any point outside the boxplot whiskers is considered an outlier. $n = 10$ Percolator runs with different random seeds for train-test splits were performed for single models, or $n = 5$ runs for combined models (MS2 and RT models used together). Source data are provided as a Source Data file.

Interestingly, we also note that HLA class I datasets benefited more from predictions than HLA class II datasets, which aligns with previous literature[14] (Fig. 4a–c, f, g, and Supplementary Data 2). This is evident across all RT and MS/MS models. A potential reason for this stems from HLA I peptides being more often singly charged compared to HLA II peptides: singly charged peptides have decreased fragmentation efficiency, causing lower performance of traditional peak counting-based scoring functions.

Our findings highlight how specialized models outperform generic ones. First, we note the differences in performance when rescoring spectra from different fragmentation methods. Distinct collision-induced dissociation (CID, resonance-type CID) and higher-energy collision dissociation (HCD, beam-type CID) models exist for Prosit and MS2PIP. However, only the HCD MS2PIP model was currently available on Koina at the time of writing, while the CID model is on the MS2PIP server at (https://iomics.ugent.be/ms2pip/19). Therefore, our comparison is limited to using the Prosit models. Marcu et al[43]. used CID to fragment precursors in their class I sample (Fig. 4a). We found that the Prosit CID model performed best here, achieving 15% more peptide identifications over baseline and outperforming both its Prosit HCD counterpart (11%) and DIA-NN (6%). Likewise, the Prosit HCD model performed better on HCD fragmentation datasets[42,43] (Fig. 4b, c). Whichever model is superior is also evident in the distribution of target PSMs' spectral similarity score distribution, which is shifted to lower similarity when the wrong fragmentation model is used (Supplementary Fig. 19).

Specialized models also improve the rescoring of MS/MS spectra from different mass spectrometers. Strangely, although Prosit timsTOF improves over DIA-NN and the Prosit HCD model trained on Orbitrap data for both class I (Prosit timsTOF: 19.1% improvement over baseline, Prosit HCD: 18%, DIA-NN: 12.8%) and II data from Phulphagar et al[45]., this does not extend to the PeptDeep models, where "Lumos" PeptDeep predictions helped identify more peptides than "timsTOF" PeptDeep predictions did (Fig. 4f, g). We believe this discrepancy may arise because PeptDeep and Prosit were trained on spectra of different NCEs. While Prosit timsTOF was trained on energies ranging from 20–70 eV, PeptDeep timsTOF was only trained on 32–52 eV[6]. More importantly, all timsTOF spectra were annotated as 30 eV during the PeptDeep training phase. Though spectra do differ across Orbitrap and timsTOF instruments, there are certain hotspots across energy levels where they are highly correlated[15,56]. Specifically, precursors fragmented with lower energies (around 20% and 20 eV on Orbitrap and timsTOF instruments, respectively) produce nearly identical spectra across instruments. Phulphagar et al. employed a scheme where precursor ion mobility and fragmentation energy were inversely correlated, with energy ranging from 55–10 eV. Though MSBooster's NCE calibration step seeks the best NCE value for spectral prediction, the lower bound of this experimental NCE range is far below that of what PeptDeep timsTOF was trained on, potentially resulting in suboptimal predictions. However, because low-energy fragmented spectra are similar between Orbitrap and timsTOF, the PeptDeep model predicting for a Lumos instrument performs better since it was trained on spectra with HCD energy as low as 20%. Indeed, when we rescored another dataset acquired on a timsTOF with similar ion mobility settings as the data PeptDeep was trained on[57], both PeptDeep and Prosit timsTOF models outperformed their Orbitrap counterparts (Supplementary Fig. 20). Koina outperformed DIA-NN's predictions the most on the CID and timsTOF datasets. This improved ability to identify peptides provides a compelling reason for incorporating the many models on Koina into FragPipe, as models trained on diverse datasets often improve upon DIA-NN's predictions.

### A heuristic approach to optimize model selection in MSBooster

We have observed empirically which models work best for phosphorylation, HLA, Astral DIA, and TMT data. However, it is outside the scope of this manuscript to exhaustively consider all types of proteomics data, and even within the same type of data, there is variability in the best-performing models. As Koina continues to grow and support more models, it may become overwhelming for users to determine which combinations of models work best for PSM rescoring in their data. To make it easier for the users to maximize peptide identifications, we have incorporated an optional module in MSBooster that attempts to determine the best MS/MS and RT model for a dataset. Importantly, the selected models may be from different frameworks (e.g., DeepLC for RT and Prosit for MS/MS), a benefit conferred by Koina's one-stop shop for all these models.

Rather than having users rescore their entire dataset using each available model, we implemented a heuristic approach that selects models based on the agreement between their predicted and experimental values of a subset of the data (Methods). When developing this algorithm, we initially chose the MS/MS model with the highest median similarity and the RT model with the lowest median RT difference but found that the algorithm would sometimes report the wrong models, according to our empirical results. Figure 5 depicts one such case for an HLA DDA class I dataset[43]. Though our algorithm correctly chose Prosit CID as the best MS/MS model (Fig. 5a), it also chose Prosit rather than DIA-NN as the best RT model (Fig. 5b). Though Prosit had the lowest median delta RT (1.38 minutes) and DIA-NN's prediction model had the largest (2.03 minutes), out of the models tested, it was actually DIA-NN's RT predictions that resulted in the most peptides identified on average, with a 6.1% increase over baseline. Compare this to Prosit RT, which performed the worst, with a 3.8% improvement over baseline and 312 fewer peptides than when using DIA-NN's predictions. Supplementary Fig. 21 shows this same analysis for the other datasets.

It was clear that the median metric often produced incorrect predictions for the RT model search, so we tested several other metrics and summarized their performance across all datasets used here (Fig. 5c-d, Methods). We found that the "top consensus" metric performed well in finding the optimal RT model. By taking the ten largest delta RT values, the algorithm could better determine the best-performing RT model than the median method. Specifically, the top 10 consensus methods achieved a "heuristic summary score" of 0.899, while the median method scored 0.793 (Methods). While the best MS/MS method was achieved by taking the bottom 100 lowest spectral similarity scores (0.981 heuristic summary score), the median similarity method performed very similarly (0.978), and the methods chose different models in only two out of the thirteen datasets. Therefore, there was insufficient justification to move away from the median method, so we continue to use it to determine the best MS/MS model. The final metrics chosen were "median" for MS2 models and "top consensus" with 10 PSMs for RT models.

We also considered PSM rescoring using all models together. We calculated a separate score for each applicable model and used them together in Percolator rescoring. Using multiple MS/MS and RT models resulted in even more peptides than when using a single model each for MS/MS and RT (Supplementary Fig. 22). Single models recovered between 42 and 86% of the total peptides added on top of the default DIA-NN's prediction by multiple model mode. This suggests that for most cases here, a single well-chosen model identified the plurality of peptides from multiple model mode without spending the time to predict all peptide candidates multiple times. A mode leveraging the predictive power of multiple Koina models simultaneously can be added to MSBooster in the future.

## Discussion

In this study, we demonstrate how Koina simplifies access to machine learning models in the proteomics domain by providing a unified platform for ML inference. Multiple popular ML models, covering different frameworks and approaches, have been implemented to

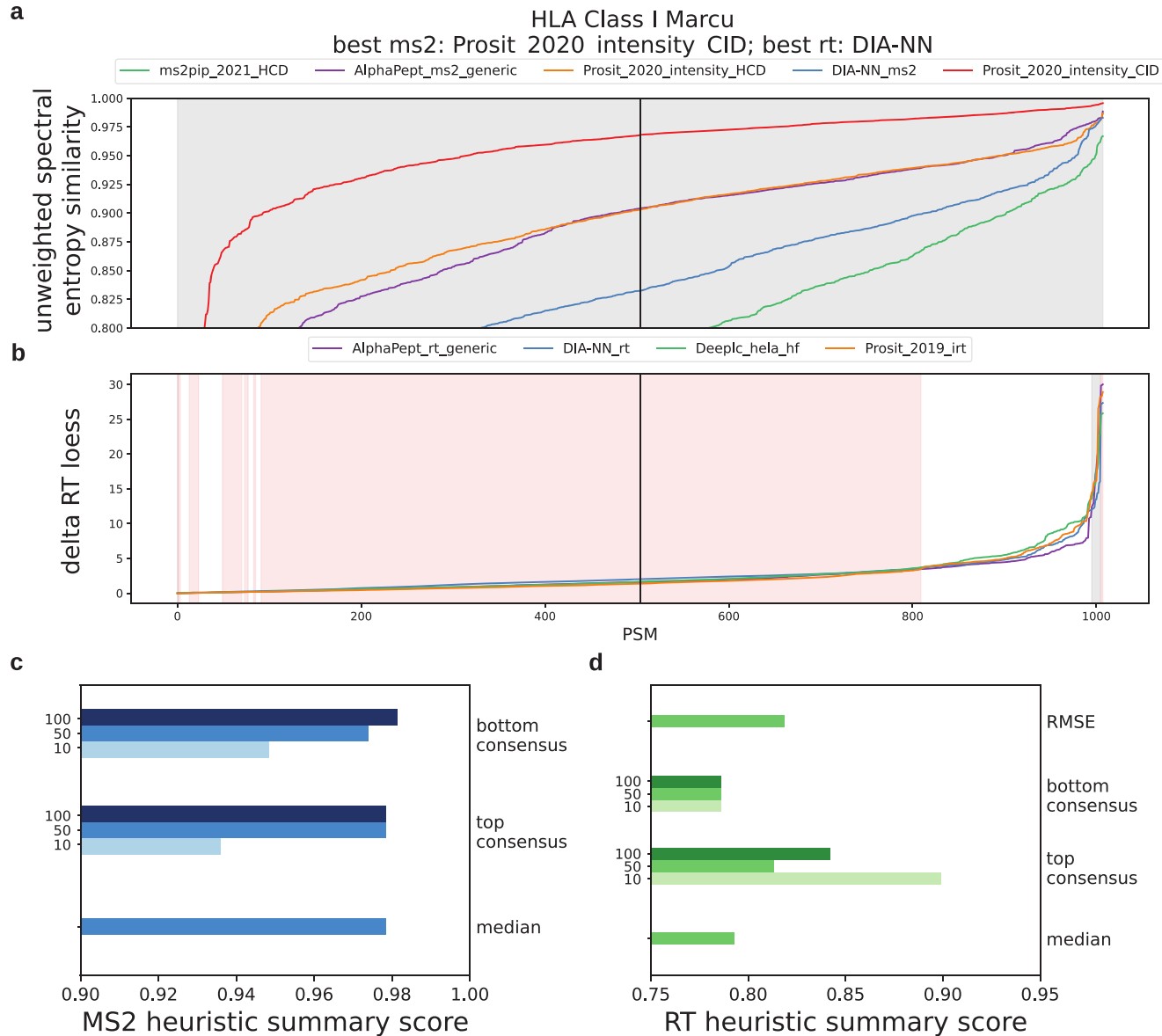

**Fig. 5 | Heuristic best model search. a** Spectral similarity and (**b**) delta RT for the top 1000 most confident PSMs according to e-value, of a HLA class I dataset[43] rescored with the predictions of multiple models. PSMs are sorted in increasing order of spectral similarity and RT scores for each model (i.e., the lowest delta RT assigned to the 1000 PSMs is plotted at position zero along the *x* axis). The black line indicates the position of the PSMs at the median of the ranked lists. Heuristic methods are based on PSM scores at certain percentiles within the ranked lists. Gray regions indicate positions at which the empirically best model for PSM rescoring also has the best spectral similarity/RT score out of all models (Fig. 5a, b). Red regions show positions where a model that performs especially worse than the empirically best model at PSM rescoring has the best spectral similarity/RT score, while white regions show where these models have comparable performance (Methods). The performance of multiple heuristic methods was summarized across all datasets for (**c**), MS/MS and (**d**) RT models. For the top and bottom consensus methods, the values of N PSMs tested were 10, 50 and 100 PSMs. Source data are provided as a Source Data file.

predict gas phase fragmentation, chromatographic retention, and collisional-cross-section of peptides. Koina extends the approach of model repositories seen in other domains by enabling users to utilize models locally while also offering web-based access. This approach merges the accessibility of a model repository with the ease of use of a web service. Combined with the freely available instances of Koina, this strategy holds significant potential to democratize machine learning, enabling laboratories with limited resources and expertise to leverage high-performance computing infrastructure for their data analysis needs. Moreover, by improving accessibility to these state-of-the-art ML models, we improve collaboration across different tools available to the scientific community by allowing both ML developers & downstream tools developers to focus on

their individual expertise while still benefiting from each other's work. We encourage ML developers to submit their models to Koina, which will not only enhance the platform's diversity but also broaden the impact and accessibility of their models. With Koina providing easy access to a continuously growing number of available models, more data analysis software will ultimately integrate with it. Finally, while fine-tuning models directly within Koina is not possible due to the platform's public and decentralized nature, an integration with an external AutoML framework could feature the direct publication of fine-tuned models on private instances. This would dramatically simplify data analysis with custom ML models, allowing a much larger number of scientists to leverage the cutting edge of ML in proteomics.

The common model interface implemented for Koina enables third-party tool developers to integrate access to any model available through Koina. While this already constitutes a major improvement regarding model interoperability, further advancements can be made. Most notably, we plan to develop an API to document model limitations in a machine-readable format. Standardizing this in the form of an API improves interoperability between models, allowing downstream data analysis tools to filter their inputs dynamically, and giving more flexibility to the user to choose the preferred level of confidence they expect of a model. Moreover, the recent release of Unispec offers a valuable opportunity for Koina to adopt a more comprehensive standard for fragment ion annotation syntax. In a future update, we will implement the recently published mzPAF standard[58] across all models, which will help to further homogenize our developed interface.

Koina's approach is entirely agnostic to the biological or technical source of the dataset; cutting-edge fields such as single-cell or spatial proteomics are natively supported. Furthermore, the platform is not inherently limited to proteomics. It can be effortlessly expanded to other domains. To demonstrate this, we have already integrated two metabolomics models that predict the tandem mass spectra fragmentation of small molecule metabolites[59]. As we continue this extension, we face the decision of whether to establish independent platforms for each domain or to centralize various models into a singular, adaptable platform. Balancing infrastructure complexity and hardware utilization will be key in this decision. A crucial development to enable a service shared across domains is an improved load-balancing mechanism that allows the assignment of models to specific instances. This will also improve performance when multiple users request predictions from different models since this currently requires the repeated loading and unloading of models, introducing additional overhead.

We illustrate how Koina's integration in FragPipe enables a benchmarking study of state-of-the-art models without having to run each model in a separate analysis pipeline, thereby isolating predictor performance from confounding factors such as database search and FDR control. We gained insights into which models perform well on specific data types and touched on important considerations for training and evaluating these models. Our results align with a recent study that evaluated strategies for training peptide-property models with scarce data[60], suggesting that single-task learning outperforms multi-task learning. While it is convenient to have a single model to handle all predictions, creating multiple specific models allows for greater flexibility in learning weights specific to the data at hand. All models considered in this study only predicted y and b ions. Recent models[21] can predict other backbone, internal, and immonium ions. Evaluating the effect of including different sets of ions has the potential to improve peptide identification, especially for non-tryptic peptides.

Anticipating the future availability of many additional models, we developed a heuristic best model search module in MSBooster. We found that the model with the highest MS/MS median similarity or lowest median RT difference did not always lead to the best PSM rescoring results. We caution readers that a model may outperform other models in popular metrics, but this does not mean it can be assumed to be the best model for PSM rescoring or for any task that relies on peptide predictions. These findings also inform the choice of loss functions used when training new prediction models. New RT models could show better performance when trained with a loss function that focuses more on overall or outlier performance. Optimizing predicted spectral libraries for PSM rescoring is a complex problem that benefits from examining ML-based feature scores for high-scoring targets and decoys, as well as those PSMs on the border of the FDR cutoff.

Finally, we have shown that using multiple models to calculate the same similarity features can further increase peptide identifications.

Our heuristic search algorithm assigns a score to each model, so it is possible to extend MSBooster in the future to calculate the same features for multiple models for a dataset. This raises an interesting question about the trade-offs regarding runtime vs the potential benefits. The prediction speed we have shown, of ~1.5 minutes for 2 million spectra using an average-performing model, indicates that even when utilizing prediction with multiple models, the introduced overhead is negligible compared to other steps in the data analysis pipeline. A potential side effect of including highly linearly correlated features may be less intuitive Percolator feature weights. Nonlinear methods such as those in Mokapot may remedy this[61].

In summary, with Koina, we create a platform for simplifying access to diverse ML models in proteomics, promoting interoperability and ease of integration. Our study highlights the strengths and limitations of existing models, providing a foundation for future enhancements. By inviting developers to contribute to Koina, we aim to foster a collaborative environment that propels advancements in machine learning for proteomics, ultimately benefiting the entire scientific community.

## Methods

### Koina Infrastructure

Koina models are hosted using Nvidia Triton, an inference server supporting ML models developed in all major prediction ML development frameworks. To unify the interface of Koina and to make ML models usable without knowledge of pre- and post-processing, Python models are used to transform inputs/outputs. Models are tied together with the ensemble model functionality, referred to as an execution graph within the context of this manuscript. Models are made available by the Kserve API implemented by Nvidia Triton Inference Server, providing access using REST and gRPC interfaces. Nginx is used to balance the load of the public Koina network between all integrated instances with a round-robin method. A Docker image is provided to simplify the deployment of new Koina instances, to minimize its size, model binary files are not stored in the image but dynamically fetched from Zenodo once the server is deployed. Both the image and the OpenAPI documentation provided at koina.wilhelmlab.org are automatically created, tested, and deployed using a custom GitHub Actions workflow.

### Data security and protection

We adhere to data protection regulations by collecting only non-identifiable data. Specifically, we gather summary statistics such as source IP addresses, content sizes and the number of predictions per request, which are necessary for preventing abuse and providing usage statistics for future grant proposals. Importantly, we do not collect the inputs users provide, and these are processed in real time without being stored on non-volatile devices.

### FragPipe analysis and database searches

All analyses were done using FragPipe 21.0 with MSFragger 4.0[32] for database searching, DIA-Umpire for pseudo-MS/MS spectra generation for HLA DIA searches[62], Percolator 3.6.4[38] for PSM rescoring, PTMProphet for phosphorylation localization[63], ProteinProphet[64] for protein assignment, Philosopher 5.1.0[33] for FDR filtering and reporting, and IonQuant 1.10.15[65] and TMT-Integrator 5.0.7[66] for TMT quantification and summarization. MSBooster 1.2.2[4] was used to turn deep-learning predictions into features before PSM rescoring, while an updated MSBooster 1.2.30 was used for timing and to demonstrate heuristic model searching. Databases for searches were downloaded within the FragPipe GUI from UniProt with reviewed Swiss-Prot sequences only, and all included common contaminants and reversed sequence decoys. The human database was downloaded on March 18, 2022 (20410 entries); mouse December 19, 2023 (25658 entries); and Arabidopsis June 2016 (the fasta available from Mergner et al., 48477 entries).

All workflows were adapted from those available in FragPipe, with specific workflows and parameters for each tool provided at (https://github.com/Nesvilab/MSBooster/tree/master/Koina%20manuscript%20resource). Data was searched using workflows adapted from those packaged with FragPipe. We relied on FragPipe headless mode (https://fragpipe.nesvilab.org/docs/tutorial_headless.html) for analyses. In silico digest produced peptides starting from length 7 to length 25 for non-specific searches or 50 for tryptic searches. 2 missed cleavages were allowed except for in Astral DIA searches, where 1 missed cleavage was allowed. 150 top intensity peaks were retained for tryptic searches, 300 for non-specific, and 500 for Astral DIA searches. In HLA searches, experimental fragment peak intensities were square-root transformed. Carbamidomethylation of cysteine was a fixed modification except for those HLA searches in which there was no alkylation step, and methionine oxidation and N-terminal acetylation were set as variable modifications. Pyro-glutamation of glutamine and glutamic acid were included as variable modifications in HLA searches. TMT searches used a fixed lysine modification and variable N-terminal and serine modifications for the TMT11 tag.

MSBooster was run on the command-line to produce output files specific to each prediction model. Version 1 of the models were tested. Critically, the "editedPin" parameter was set to an empty string so that the next processing steps would recognize the pin files. Percolator was run with "--subset-max-train 500000" for Astral DIA data only. Results were filtered with Philosopher at 1% FDR at the PSM and peptide levels. 1% protein FDR filtering was also applied for all searches besides HLA searches.

### Throughput and latency benchmarks

Two servers (each: Intel(R) Xeon(R) CPU E5-2640 v4 @ 2.40 GHz; 4x GTX 1080; 64GB RAM) were temporarily removed from the Koina network and dedicated to this benchmark. On each server four instances of Koina were available. The Triton Performance Analyzer was used to benchmark models consecutively over a range of client-side concurrency (2, 4, 6, 8, 10, 20, 40, 60, 80, 100), number of utilized Koina instances (1, 2, 4, 8) and two distinct geographic locations (Ann Arbor, Michigan, USA and Freising, Germany) recording throughput (infer/sec) and latency (microseconds).

### MSBooster speed benchmarks

Timing was recorded on two computers: (1) Linux server: Intel(R) Xeon(R) Gold 6354, 3.00 GHz, 36 cores, 72 logical processors and 755GB RAM; (2) Windows laptop: Intel(R) Core(TM) i7-9750H, 2.60 GHz, 6 cores, 12 logical processors and 16GB RAM. Each dataset was predicted ten times using multithreading with 55 threads as specified in the MSBooster parameter file, and the average timing per 1000 peptides is shown in Fig. 3c. The maximum heap space was set with the Java -Xmx parameter, either as 256GB on the Linux server or 8GB on the Windows laptop.

### NCE calibration in MSBooster

The top 1000 PSMs ranked by expectation value (e-value) across all pin files are extracted and predicted at all integer values in a certain range (20–40%, by default). Predicted and experimental spectra are compared with the unweighted spectral entropy similarity metric[4,67] and the NCE value that produces the highest median similarity is selected when calling the model to predict all other peptide candidates from the pin files.

### Heuristic best model search scoring

Multiple methods were developed to approximate the best model for a dataset without having to rescore all PSMs and process them through downstream tools. First, the $1000/P$ PSMs with the lowest expectation values (as calculated by MSFragger using the database search hyperscore[32]) are selected from each pin file, where $P$ is the number of

pin files. Next, all PSMs are sorted in ascending order of their unweighted spectral entropy (MS/MS) or delta RT loess (RT) score. Then, each heuristic method was applied. Loess alignment is applied to mitigate the effects of the experimental setup on the predictions.

- Median: The median score calculated using the predictions of each model is determined. The MS/MS model with highest median similarity and RT model with lowest median RT difference are chosen as the best models.
- Top consensus: N PSMs with the largest values are selected and sorted. Largest values for the PSM features when testing MS/MS models mean those with the highest spectral similarity; when testing RT models, large values mean the greatest deviations from the RT calibration curve. The values of N tested are 10, 50 and 100. At every position from 1 to N, the model with the best PSM feature value (greatest MS/MS similarity or smallest RT deviation) gets a vote. The model with the most votes was selected. The theoretical bounds for this are 0 when a model gets no votes or 1 when a model gets all votes.
- Bottom consensus: This method is similar to the top consensus method, but it focuses on the PSMs with lowest MS/MS similarity and smallest RT deviation.
- RMSE: The root mean squared error of RT deviations.

Once each heuristic method had produced its pick for best model, the average number of peptides identified empirically across 10 runs with different Percolator random seeds was divided by the average peptides identified by the model with the highest average. This was applied separately to RT and MS/MS models. The square of this ratio was averaged across multiple datasets to produce the final heuristic summary score shown in Fig. 5c, d. The average squared ratio value, though less intuitive than an average ratio value, was better suited for our need to apply a heavier penalty to smaller ratios. For phosphoproteomics datasets, all peptides were considered, not only phosphopeptides. For TMT datasets, all peptides were considered, not just those quantified.

To determine the regions of gray, red, and white in Fig. 5a, b the model with best spectral similarity/RT score was recorded at each position of the sorted lists. If the model with the best score at a PSM position was also the best PSM rescoring model, this position is assigned the color gray. If the model with the best score at a position does not match the best PSM rescoring model, the position will either be colored white or red. Red indicates that the mean peptides identified by this model plus one standard deviation is less than the mean peptides for the empirically best model minus one standard deviation, based on results listed in Supplementary Data 2. White indicates that the best model at the position and best PSM rescoring model have comparable performance (i.e., the mean peptides for this model plus one standard deviation is greater than or equal to the mean peptides for the empirically best model minus one standard deviation).

### Reporting summary

Further information on research design is available in the Nature Portfolio Reporting Summary linked to this article.

## Data availability

Unless otherwise stated, all data supporting the results of this study can be found in the article, supplementary, and source data files. The source data generated in this study has been deposited in the Zenodo database under accession code 16754391. Raw MS Data for the FragPipe analysis was downloaded from the ProteomeXchange Consortium with accession codes PXD013868, PXD030983, PXD020186, PXD022950, PXD010012, PXD049028, PXD046372 and PXD036025, and from the MassIVE repository with codes MSV000081439 and MSV000091456. Source Data are provided with this paper.

## Code availability

The source code for Koina and the Python client package is available on GitHub as wilhelm-lab/koina and archived on Zenodo under accession code 16800964. The source code for the R client package is available at Bioconductor as koinar (https://bioconductor.org/packages/koinar/). Source code for MSBooster is available on Nesvilab/MSBooster (https://github.com/Nesvilab/MSBooster). FragPipe is available for download at https://fragpipe.nesvilab.org/. Source code and downloadable binaries for EncyclopeDIA are available on Bitbucket as searleb/encyclopedia (https://bitbucket.org/searleb/encyclopedia). Source code for Skyline is available on GitHub as ProteoWizard/pwiz (https://github.com/ProteoWizard/pwiz) an installer is available at (https://skyline.ms/). Source code for Oktoberfest is available on GitHub as wilhelm-lab/oktoberfest (https://github.com/wilhelm-lab/oktoberfest).

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

## Acknowledgements

We thank the FGCZ ETHZ|UZH for hosting a Koina instance and Marco Schmidt for configuring the networking. Furthermore, we would like to thank the organizers of the EuBIC-MS - Developers Meeting 2023 for giving us a chance to start this project there. We thank Naim Abdul-Khalek and Simon Gregersen Echers for their support to integrate the Pfly model in Koina. We thank Dario Strbenac and Vincent Carey for their constructive code reviews of the KoinaR Bioconductor package. We also thank all Wilhelm and Nesvizhskii Labs members for their valuable input and feedback. This research was in part funded by the Munich Data Science Institute (L.L.); European Union's Horizon 2020 Program under Grant Agreement 823839 [H2020-INFRAIA-2018-1; EPIC-XS] (W.G.); an ERC Starting Grant [101077037] (E.K., M.W., W.G.); National Institutes of Health grants R01-GM094231, 1U01CA288888, and U24-CA271037 (K.L.Y., F.Y., K.L., A.I.N.); R35-GM150723 (B.C.S.); R35-GM150583 and R00-HD090201 (D.B.W.); the University of Michigan Rackham Pre-doctoral Fellowship (K.L.Y.); the Research Foundation Flanders (FWO) [12B7123N, 12A6L24N, 1SH9O24N] (R.G., R.B., R.D.); National Science Foundation (NSF) Grant No. DBI-2011271 (H.T., Y.H.); the Spanish Ministry of Science and Innovation [PID2020-115092GB-I00] (E.S.); and NIH grants 5U24DK133658-02 (Mi.W., M.S).

## Author contributions

L.L. developed Koina and the complementary Python and Bioconductor packages. K.L.Y. and F.Y. developed the integration with FragPipe and performed the benchmarking of different models using FragPipe. D.B. developed a prototype of Koina. L.L. & E.K. developed the documentation webpage. M.C. and B.M. developed the integration with Skyline. K.L. added support for Koina predictions in FragPipe-PDV. B.C.S. developed the integration with EncyclopeDIA. L.L., T.K., C.P., W.G., T.S., D.B.W., M.S., Y.H., H.T., Mi.W., R.G., R.B., R.D., J.A., E.S. contributed models to Koina. T.S. conceived the approach. A.N., M.W. supervised the research, providing guidance throughout the project. L.L. and K.L.Y. wrote the manuscript and prepared visualizations. All authors reviewed, edited and approved the manuscript.

## Funding

## Competing interests

A.I.N. is the Founder of Fragmatics and serves on the scientific advisory boards of Protai Bio, Infinitopes, and Mobilion Systems. A.I.N. is also a paid consultant for Novartis. A.I.N., F.Y., and K.L. have a financial interest due to the licensing of MSFragger, IonQuant, and diaTracer to commercial entities. Mi.W. is a co-founder of Ometa Labs LLC. B.C.S. is a founder and shareholder in Proteome Software, which operates in the field of proteomics. The Searle Lab at the Ohio State University has a sponsored research agreement with Thermo Fisher Scientific. T.S. is a co-founder, shareholder, and employee of MSAID GmbH, a company that develops software for proteomics. M.W. is a co-founder and shareholder of MSAID GmbH and a scientific advisor of Momentum Biotechnologies, but he has no operational role in either company. The other authors have no competing interests to declare.
