## [Transparent Peer review file · Nature Communications]

Koina: Democratizing machine learning for proteomics research

Corresponding Author: Professor Mathias Wilhelm

Version 0:

Reviewer comments:

Reviewer #1

(Remarks to the Author)

In this study, Koina was developed as an open-source containerized, decentralized, online-accessible, and extensible high-performance model repository. However, the novelty of this study is limited and some serious problems are needed to be resolved.

1. The differences of these ML/DL models should be provided in the Koina. Users can select the corresponding model based on these differences.
2. The novelty and significance of this article are not sufficiently highlighted in the Introduction section, and only construction of model-specific web servers based on the existing tools is limited for the novelty.
3. Only one case is not enough for benchmarking Koina. More different types of data should be used to test this tool.
4. The website is not user-friendly and lacks a section that explains the meaning of each button and the output results.
5. If the input data is very large, how does this tool handle it?
6. In addition to the models predicting peptide properties used in computational proteomics analysis, there are preprocessing and post-processing steps. How can users optimize the preprocessing and post-processing steps?
7. In benchmark dataset, how can peptide identification for specific datasets be optimized to account for the differences in charge state and fragmentation efficiency between HLA I and HLA II peptides?
8. Can the impact of different mass spectrometers and fragmentation methods on the performance of prediction models be mitigated through a unified cross-instrument data training framework?
9. What are the potential benefits and computational trade-offs of incorporating multiple MS/MS and RT model predictions simultaneously in PSM rescoring, compared to single-model approaches?

(Remarks on code availability)

The code provide a README file with enough instructions for installing and running the application

Reviewer #2

(Remarks to the Author)

The authors present Koina, a server which aims to provide uniform and accessible use of machine learning models for predicting peptide properties. Use of this server is already integrated into several proteomics tools, and the authors provide an in-depth look at its use and benefits as a new feature of FragPipe.

The manuscript is well written. With minor exceptions noted below, all information is presented clearly and will likely be accessible to computational experts and non-experts alike.

The presented tool, Koina, is a relevant and important tool for proteomics research. Having worked quite a bit with various proteomics-related ML models in the past it is nice to see this work being done. Comparing models from different "ecosystems" has always been a large frustration and, as the authors suggest, this almost certainly leads to many groups simply using whatever is most familiar rather than what is most suitable.

We would recommend publishing this manuscript after the following comments are addressed.

Comments:

Page 6:

The authors mention that the online documentation is semi-automatically generated and that "This means that ML developers adding new models to Koina do not need to be familiar with web development to provide easily accessible documentation for their models". This is good, but I think it would be very helpful for there to also be manual annotation (preferably by the Koina team to ensure consistency and correctness of what is documented on their website) of the training data and any limitations of the model. This could be a simple table for each model with the range of peptide lengths and variable and fixed modifications used in the training, plus any other relevant things like which ions are predicted (i.e. neutral loss ions? Internal fragments?) Supplementary Table S1 provides some limited information in that regard but this information (e.g. on peptide length limitations) is absent on the Koina website, even though it is crucial for understanding the suitability of models for a given task.

For example, the first model in the list on the "Try it out" section is DeepLC_hela_hf. It states "The DeepLC HeLa HF model was trained from random parameters on 161 193 tryptic peptides. Modifications for this model include carbamidomethyl, oxidation of methionine, and N-terminal acetylation." We don't know the relevant peptide length range. While it might be assumed that carbamidomethyl is a fixed modification, we do not know. How are we to encode this modification? Is it implied in every C or does it require a mod label?

The second model is AlphaPeptDeep_rt_generic. It quite simply states: "Find out more about this model here", with a link to the GitHub repository. The Mann Lab does have good GitHub repositories, but the linked page is just the AlphaPeptDeep readme and does not tell us anything about the scope of application and limitations of the rt_generic model.

Page 8, Line 174:

Table S1 states ProSIT does not support peptides of length > 30. However, it is therefore unclear to us why the following code testing prediction for a 36-mer peptide does not raise any errors:

```
curl "https://koina.wilhelmlab.org:443/v2/models/ProSIT_2020_intensity_HCD/infer" -d '{
  "id": "0", "inputs": [
    {"name": "peptide_sequences", "shape": [1,1], "datatype": "BYTES", "data":
    ["PEPTIDEPEPTIDEPEPTIDEPEPTIDEPEPTIDER"]},
    {"name": "precursor_charges", "shape": [1,1], "datatype": "INT32", "data": [1]},
    {"name": "collision_energies", "shape": [1,1], "datatype": "FP32", "data": [25]}
  ]
}'
```

Page 9, Figure 1:

Koina standard output is referenced but it did not become clear to us what this enforces beyond a tabular format. Please discuss the limitations of this standardization:

E.g., the annotation column seems to be free text, where for example the unispec model outputs unfamiliar internal fragment descriptions. Also, most models seem to output spectra normalized such that $\max(\text{intensity}) = 1$. However, ms2pip models output spectra with $\sum(\text{intensity}) = 1$.

Page 11, paragraph 2:

Please clarify exactly how the comparison between DIA-NN and Koina in FragPipe is being done. In particular, it was not clear if Koina is being called from a local server or if the default remote server is being used.

Page 17:

The method chosen for selecting the best RT model for an experiment is described as follows: Predictions from all models are made for the top 1,000 PSMs, including both target and decoy PSMs. The algorithm then takes the ten largest delta RT values to determine the best performing RT model.

First: how are the top 1,000 PSMs selected? Is this done after Percolator rescoring or is it based on the raw MSFragger output? What is the metric being used?

Second: We assume that the model with the smallest top-ten delta RTs is the one which is selected, but this is not explicitly stated (please clarify). However, this seems strange because the top ten delta RTs will be enriched for decoys. How is this handled? Is it intended?

Minimizing the top ten delta RTs of the targets makes sense for selecting the model with the tightest distribution, but ideally

the delta RTs of the decoys are, to an extent, more or less random and would not be meaningful for optimizing for the targets. Are decoys actually used in the model selection or are they excluded?

Page 20, Figure 5:

Panel a) is described to display spectral similarity. In our understanding, the y axis would then have to be “unweighted (spectral) entropy similarity”, as “spectral entropy” is not a measure for comparing spectra.

Panel b): Please clarify whether the x-axis sorted (e.g. “PSM rank”).

Panel a-b) It is not clear to us why Panel a) directly uses an attribute of the PSM (entropy similarity) whereas for the delta RT axis, a loess smoothing had to be employed.

Page 22, Lines 586:

It is appropriately stressed that models scoring best in one metric might not perform best in terms of overall IDs. We would consider it helpful to also discuss that koina models may predict different kinds of ions which may factor into this performance in a way that was not evaluated in the manuscript. E.g. the unispec model predicts many more ions than prosit, and inclusion/exclusion of which affects the spectral similarity calculations.

(Remarks on code availability)

The code is understandable. The online interface is user-friendly and functional, and we were able to easily spin up a local server for testing.

The online documentation is easy to read and understand and is generally good. There is some room for improvement in the description of model restrictions, training data, and output. Comments regarding these points are in the main "Comments for author" section.

Reviewer #3

(Remarks to the Author)

The authors present a platform, Koina, designed to enhance accessibility and usability of machine learning or deep learning (ML/DL) models for proteomics research. They aim to lower barriers for laboratories with limited computational resources, facilitating the application of state-of-the-art ML/DL tools in MS-based analyses. While the manuscript addresses a critical need in the proteomics community, certain areas require further elaboration and refinement to meet the standards of Nature Communications.

Major issues:

(1) The development of Koina as a containerized, decentralized, and web-accessible ML platform is a commendable effort to democratize proteomics research. However, the manuscript would benefit from a clearer articulation of Koina's unique contributions compared to existing platforms, such as BioModels. A side-by-side comparison of functionality, ease of use, and performance metrics would enhance the clarity and impact of the work. Also, some case studies are highly recommended to show Koina's unique features. The user case of HLA data analysis actually shows the advantages of the PSM rescoring model, rather than Koina itself.

(2) Some technical implementation requires more detailed explanation. Providing these details would enhance confidence in the platform's high performance.

- How does Koina ensure scalability and low latency under high user demand?
- What are the benchmarks for server response times, throughput, and model execution speeds under varying conditions?
- How does Koina's performance compare when running the same models locally versus via the platform?

(3) I think the heuristic approach for selecting optimal models (e.g., for retention time or MS2 predictions) is practical but limited. The authors are encouraged to explore the integration of Automated Machine Learning (AutoML) into Koina. Currently, the results in this work seem to suggest that there is no objective metric for selecting optimal models.

(4) Integration with popular tools like FragPipe and Skyline is a major strength. Koina provides more options for model selection. However, the manuscript should provide more detailed examples or case studies that demonstrate the practical benefits of these integrations in real-world scenarios. Did this flexibility of Koina lead some advantages (such as performance improvement) for end users? Currently, I could not find a clear advantage for bioinformaticians who have programming skills. And for biologists who have few programming skills, both the standalone tools and Koina are hard to use for themselves. More detailed tutorials (even including user-guide videos) are needed.

(5) Another important issue is the authors should make a detailed and clear description about the governance of a community-driven platform (i.e., Koina), including mechanisms for quality control, validation, and versioning of models submitted to Koina.

(6) Meanwhile, the decentralized architecture of Koina raises important questions about data security and privacy, particularly for sensitive clinical datasets. The authors should explicitly address:

- How data security is maintained during processing.
- How did they comply with data protection regulations.

(7) The manuscript's discussion of future development is limited. A more detailed roadmap, including the addition of AutoML features or other enhancements (e.g., instrument-specific models), would strengthen the paper's vision for Koina's growth.

Minor issues:

1. In practical analysis tasks, users may need to fine-tune existing models, and some models (eg. PeptDeep) also support fine-tuning and then prediction tasks. So, does Koina support fine tuning? I didn't see that in the manuscript and Koina's

documents.

2. What is the definition of "top consensus" metric in Fig.5?

(Remarks on code availability)

(1) I were able to install the docker image of Koina on a Linux server. But I suggest considering the compatibility with Windows.

(2) More detailed tutorials (even including user-guide videos) are needed.

Reviewer #4

(Remarks to the Author)

(Remarks on code availability)

Version 1:

Reviewer comments:

Reviewer #1

(Remarks to the Author)

The authors have addressed most of the concerns raised in the previous review, and the manuscript has been significantly improved. However, a few issues remain unresolved and should be clarified before publication:

1. While the platform's performance has been demonstrated using synthetic or simulated datasets, its utility must be validated with real-world proteomics data. The authors are suggested to apply their method to experimentally derived protein datasets (e.g., from mass spectrometry or antibody-based assays) to identify bona fide proteins and perform downstream functional analyses (e.g., pathway enrichment, protein-protein interactions). This step is essential to demonstrate practical relevance and robustness.

2. The manuscript would benefit from a broader perspective on the platform's applicability to cutting-edge fields such as single-cell proteomics and spatial metabolomics.

(Remarks on code availability)

Reviewer #2

(Remarks to the Author)

The authors have addressed all concerns. We have one minor comment, explained below. This comment is not blocking publication in our opinion, and the authors can consider if they want to address it when making final edits/revisions.

Page 20, Figure 5:

The y-axis of Figure 5a is labeled "unweighted spectral entropy". We made a comment about this last time but it might have been misinterpreted. We think the label should be "unweighted spectral entropy similarity", the "similarity" part being important because the score is comparing two different spectra. "Entropy" alone would describe the information content of a single spectrum, not the similarity of two spectra. However, it is possible that the use of plain "entropy" as the name of the metric is common in the literature, so we leave it up to the authors to decide how to label the figure.

(Remarks on code availability)

The code is understandable. The online interface is user-friendly and functional, and we were able to easily spin up a local server for testing.

The online documentation is easy to read and understand. The additions the authors have made to address previous comments are impressive and helpful.

Reviewer #3

(Remarks to the Author)

I appreciate that the authors have addressed most of my previous concerns. Here, I only have one minor issue to be considered: What are the theoretical bounds for "top consensus"? Can the authors provide a preferred value?

(Remarks on code availability)

The codes are understandable. The tutorial files are revised and easy to follow.

Reviewer #4

(Remarks to the Author)

(Remarks on code availability)

Version 2:

Reviewer comments:

Reviewer #1

(Remarks to the Author)

The authors have addressed the concerns raised in the previous review. I have no additional comments at this time.

(Remarks on code availability)

The codes are usable

Reviewer #3

(Remarks to the Author)

I have no further questions.

(Remarks on code availability)

The codes are understandable. The tutorial files are revised and easy to follow.

REVIEWER COMMENTS

Reviewer #1 (Remarks to the Author):

In this study, Koina was developed as an open-source containerized, decentralized, online-accessible, and extensible high-performance model repository. However, the novelty of this study is limited and some serious problems are needed to be resolved.

1. The differences of these ML/DL models should be provided in the Koina. Users can select the corresponding model based on these differences.

We agree with the concerns raised, we reworked the summary documentation for all models available on Koina on our website (<https://koina.wilhelmlab.org/docs>). To further homogenize the documentation of current and future models, we constructed a questionnaire (<https://github.com/wilhelm-lab/koina/blob/main/docs/DOME.md>). This gives the documentation additional structure which helps users to quickly evaluate the use cases and limitations of available models.

2. The novelty and significance of this article are not sufficiently highlighted in the Introduction section, and only construction of model-specific web servers based on the existing tools is limited for the novelty.

We reworked the introduction to clarify the novelty of Koina. To the best of our knowledge, Koina is the only platform available right now that enables the usability (findability, accessibility, interoperability, and reproducibility) of machine learning models across domains and models - explicitly not being “model-specific” and more than just a web server. In total, we have integrated >30 models from 5 domains developed by >5 independent research groups. To highlight the unique aspect of Koina further, we added a comparison to the Supplemental Information with two other platforms (BioModelsML, Kipoi) that aim to solve related but different challenges in ML.

We also reworked the discussion to highlight the impact we believe Koina is going to have on ML and, specifically the field of proteomics.

3. Only one case is not enough for benchmarking Koina. More different types of data should be used to test this tool.

Beyond the already discussed integration in MSBooster, which has been used to benchmark the available ML models on multiple datasets comprising different mass spectrometers and different MS methods, we have now included additional benchmarks on phosphoproteomics, DIA (Astral), and TMT data in the Supplemental Information (Supplemental Fig. 15-17), which shows similar results to our results on HLA data. We hope this alleviates the concerns of the reviewer, although we are obliged to mention that the novelty of Koina is not tied to how well a specific model performs. It is rather tied to providing an easy-to-use and highly-performant platform to improve model accessibility for developers and end users.

4. The website is not user-friendly and lacks a section that explains the meaning of each button and the output results.

We reworked the website design with the aim of improving its accessibility and clarity for users. Specifically, we now provide a detailed tour for new users that goes step by step over the functionality of the website. We also restructured the website to a more focused design for the model documentation.

5. If the input data is very large, how does this tool handle it?

We agree with the concern of the reviewer that the execution time of the models is a crucial aspect when ML is applied to MS-based proteomics since the model will likely need to generate thousands to millions of predictions for a single user. The performance of Koina in this regard is more than satisfactory. Predicting spectra for a full digest of the human proteome (~2 million peptides) takes ~1.5 minutes (AlphaPeptDeep MS2). This is comparable to the performance of running the same model on a single Nvidia GTX 1080, but it alleviates (end) users from having GPUs locally (plus maintaining the hardware and software), while having access to >30 models by a press of a button. In contrast, running the same model on CPU takes much longer. We also created an in-depth performance benchmark for other models available on Koina that investigate throughput and latency under various conditions (client-side concurrency, available Koina instances, geographic location, and sequence length). The results have been summarized in new Supplementary Figures 3-6.

6. In addition to the models predicting peptide properties used in computational proteomics analysis, there are preprocessing and post-processing steps. How can users optimize the preprocessing and post-processing steps?

We believe there to be a misunderstanding about the purpose of pre- and post-processing steps. This does not refer to pre- and post-processing of spectra (noise reduction, peak detection, normalization, deisotoping, etc.). It refers to pre- and post-processing of the inputs and outputs of the respective ML model. These include, but are not limited to, aspects such as converting the sequence from the developed inter-model shared string sequence format to the model-specific integer representation of the sequence (Fig. 2). In this context, optimization would refer to execution speed, which does not need to be performed by the user because Koina already performs extremely well. This is exemplified in the newly provided benchmark assessing the throughput of all models, which has been summarized in the new Supplemental Figures 3-6. We also clarified the purpose of pre-and post-processing steps in the Results section *“We have created new pre-processing steps that convert the standard ProForma 2.0 sequence format into the specific formats needed by different models.”*

7. In benchmark dataset, how can peptide identification for specific datasets be optimized to account for the differences in charge state and fragmentation efficiency between HLA I and HLA II peptides?

It is not clear to us how this question relates to the manuscript, as the examples of rescoring shown in the manuscript are meant to show the ease of use of Koina and the plethora of

applications supported by its development. If our response is not to your satisfaction, please clarify this further.

We agree that ML models do have varying prediction performance for different peptide classes, instruments and acquisition techniques. This is one of the main reasons that led to the development of such an abundance of ML models which in turn requires the here developed service so that end users can pick the optimal model for their application. With Koina, a comparison of this performance is extremely easy to perform. With the “best model search” and collision energy calibration implemented in Fragpipe/MSBooster this can even be done automatically. If users do not want to use Fragpipe/MSBooster, Koina still dramatically simplifies this process by enabling access to a plethora of ML models that users would have to install and maintain locally otherwise. From personal experience, this can take weeks per single model. Additionally, Koina provides documentation for all of these models, condensing the relevant information of the associated manuscript dramatically.

8. Can the impact of different mass spectrometers and fragmentation methods on the performance of prediction models be mitigated through a unified cross-instrument data training framework?

Yes, it can. But the development of such a model is not the scope/purpose of the manuscript. We (the lab of Mathias Wilhelm) are actively working on such a model. Until this model is ready, Koina provides the next best alternative by allowing seamless integration of models covering various instruments and acquisition techniques - e.g. currently available models support various fragmentation methods and mass analyzers. Koina effectively increases the requirements for the novelty of such a model because the majority of the benefits such a model would bring are already covered by Koina. But even once the development of this model is finalized, models developed in our groups will be published on Koina. Giving all downstream tools that are already interfacing with Koina immediate access to them. We understand Koina as a platform to publish models to improve model accessibility and are not concerned with the performance of single models.

9. What are the potential benefits and computational trade-offs of incorporating multiple MS/MS and RT model predictions simultaneously in PSM rescoring, compared to single-model approaches?

As described in the section “Automated heuristic model search accurately determines the best model combination” we found that “for most cases here, a single well-chosen model identified the plurality of peptides from multiple model[s]”. Evaluating this comprehensively requires more time and the implementation of new pipelines, which we believe are beyond the scope of this manuscript. However, we expanded the discussion section highlighting that the computations trade-offs should not be a concern given the new performance results we show.

“This raises an interesting question about the trade-offs regarding runtime vs the potential benefits. The prediction speed we have shown, of ~1.5 minutes for 2 million spectra using an average-performing model, indicates that even when utilizing prediction with multiple models, the introduced overhead is negligible compared to other steps in the data analysis pipeline.”

Reviewer #1 (Remarks on code availability):

The code provide a README file with enough instructions for installing and running the application

Reviewer #2 (Remarks to the Author):

The authors present Koina, a server which aims to provide uniform and accessible use of machine learning models for predicting peptide properties. Use of this server is already integrated into several proteomics tools, and the authors provide an in-depth look at its use and benefits as a new feature of FragPipe.

The manuscript is well written. With minor exceptions noted below, all information is presented clearly and will likely be accessible to computational experts and non-experts alike.

The presented tool, Koina, is a relevant and important tool for proteomics research. Having worked quite a bit with various proteomics-related ML models in the past it is nice to see this work being done. Comparing models from different "ecosystems" has always been a large frustration and, as the authors suggest, this almost certainly leads to many groups simply using whatever is most familiar rather than what is most suitable.

We would recommend publishing this manuscript after the following comments are addressed.

Thank you for the positive feedback!

Comments:

Page 6:

The authors mention that the online documentation is semi-automatically generated and that "This means that ML developers adding new models to Koina do not need to be familiar with web development to provide easily accessible documentation for their models". This is good, but I think it would be very helpful for there to also be manual annotation (preferably by the Koina team to ensure consistency and correctness of what is documented on their website) of the training data and any limitations of the model. This could be a simple table for each model with the range of peptide lengths and variable and fixed modifications used in the training, plus any other relevant things like which ions are predicted (i.e. neutral loss ions? Internal fragments?) Supplementary Table S1 provides some limited information in that regard but this information (e.g. on peptide length limitations) is absent on the Koina website, even though it is crucial for understanding the suitability of models for a given task.

For example, the first model in the list on the "Try it out" section is Deeplc_hela_hf. It states "The DeepLC HeLa HF model was trained from random parameters on 161 193 tryptic peptides. Modifications for this model include carbamidomethyl, oxidation of methionine, and N-terminal acetylation." We don't know the relevant peptide length range. While it might be assumed that carbamidomethyl is a fixed modification, we do not know. How are we to encode this modification? Is it implied in every C or does it require a mod label?

The second model is AlphaPeptDeep_rt_generic. It quite simply states: "Find out more about this model here", with a link to the GitHub repository. The Mann Lab does have good GitHub repositories, but the linked page is just the AlphaPeptDeep readme and does not tell us anything about the scope of application and limitations of the rt_generic model.

We agree with the points raised, and we now provide summary documentation for all published models available on Koina that address the points raised by the reviewer (e.g., peptide length limitations, and handling of carbamidomethyl). To additionally further improve the documentation of current & future models, we constructed a questionnaire to guide model documentation (<https://github.com/wilhelm-lab/koina/blob/main/docs/DOME.md>). This should help users to more quickly evaluate use cases and limitations of the available models.

Page 8, Line 174:

Table S1 states Prosit does not support peptides of length > 30. However, it is therefore unclear to us why the following code testing prediction for a 36-mer peptide does not raise any errors:

```
curl "https://koina.wilhelmlab.org:443/v2/models/Prosit_2020_intensity_HCD/infer" -d '{
  "id": "0", "inputs": [
    {"name": "peptide_sequences", "shape": [1,1], "datatype": "BYTES", "data":
    ["PEPTIDEPEPTIDEPEPTIDEPEPTIDEPEPTIDER"]},
    {"name": "precursor_charges", "shape": [1,1], "datatype": "INT32", "data": [1]},
    {"name": "collision_energies", "shape": [1,1], "datatype": "FP32", "data": [25]}
  ]
}
```

Thank you for noticing this. We corrected this issue, Prosit models are now accepting only peptides of length <=30.

Page 9, Figure 1:

Koina standard output is referenced but it did not become clear to us what this enforces beyond a tabular format. Please discuss the limitations of this standardization:

E.g., the annotation column seems to be free text, where for example the unispec model outputs unfamiliar internal fragment descriptions. Also, most models seem to output spectra normalized such that $\max(\text{intensity}) = 1$. However, ms2pip models output spectra with $\sum(\text{intensity}) = 1$.

Thank you for the comment, we agree with the concerns of the reviewer. We extended the results section with a more detailed description of the standardization of the inputs/output format.

“To achieve this, we first standardize the data types and formats for both inputs and outputs. Next, we focus on standardizing input formats; while this is generally easy for most data types, peptide sequences require special attention. We have created new pre- and post-processing tools that convert the standard ProForma 2.0 sequence format into the specific formats needed by different models. For outputs, we aim to keep the original format to allow users to check their predictions against their original data sources. We expect this approach will have a minimal effect on interoperability, as most metrics used in later data analysis are not significantly impacted by differences in spectrum normalization or retention times”

We see the greatest limitation to be how model limitations are effectively communicated, we outlined in the discussion that we aim to develop an API to standardize this.

“[...] we plan to develop an API to document model limitations in a machine-readable format. Standardizing this in the form of an API improves interoperability between models, allowing downstream data analysis tools to filter their inputs dynamically, giving more flexibility to the user to choose the preferred level of confidence they expect of a model.”

We also note in the discussion that with the recent addition of Unispec fragment ion annotation syntax between models is not unified anymore. This was necessary because the previously used syntax did not support the variety of peaks Unispec predicts. As a future development, we will change the annotation syntax for all models to the newly developed standard mzPAF.

“Moreover, the recent release of Unispec offers a valuable opportunity for Koina to adopt a more comprehensive standard for fragment ion annotation syntax. In a future update we will implement the recently published mzPAF standard, across all models, which will help to further homogenize our developed interface.”

We also clarified the fragment ion annotation syntax in the documentation of Unispec.

Page 11, paragraph 2:

Please clarify exactly how the comparison between DIA-NN and Koina in FragPipe is being done. In particular, it was not clear if Koina is being called from a local server or if the default remote server is being used.

To measure the time used for DIA-NN's predictions, we extracted the timestamps from its printed output to the terminal and added the times for the prediction and decoding steps. To measure the time used for Koina models' prediction, we used the cumulative timing reported by MSBooster for sending the request, receiving the response, and parsing the results from JSON format to data structures usable by MSBooster.

We used the default remote server at <https://koina.wilhelmlab.org:443/v2/models/> for Koina predictions. This has been clarified in the manuscript:

“MSBooster can query and parse predictions from Koina for thousands of peptides per second using the default public server at <https://koina.wilhelmlab.org/v2/models/> (Fig. 3c, Supplementary Data S1, Methods).”

Details of the hardware and parameters used for benchmarking can be found in the Methods section under “MSBooster speed benchmarks”:

“Timing was recorded on two computers: (1) Linux server: Intel(R) Xeon(R) Gold 6354, 3.00 GHz, 36 cores, 72 logical processors, and 755GB RAM; (2) Windows laptop: Intel(R) Core(TM) i7-9750H, 2.60 GHz, 6 cores, 12 logical processors, and 16GB RAM. Each dataset was predicted ten times using multithreading with 55 threads as specified in the MSBooster parameter file, and the average timing per 1000 peptides is shown in Figure 3c. The maximum heap space was set with the Java -Xmx parameter, either as 256GB on the Linux server or 8GB on the Windows laptop.”

Page 17:

The method chosen for selected the best RT model for an experiment is described as follows: Predictions from all models are made for the top 1,000 PSMs, including both target and decoy PSMs. The algorithm then takes the ten largest delta RT values to determine the best performing RT model.

First: how are the top 1,000 PSMs selected? Is this done after Percolator rescoring or is it based on the raw MSFragger output? What is the metric being used?

Second: We assume that the model with the smallest top-ten delta RTs is the one which is selected, but this is not explicitly stated (please clarify). However, this seems strange because the top ten delta RTs will be enriched for decoys. How is this handled? Is it intended?

Minimizing the top ten delta RTs of the targets makes sense for selecting the model with the tightest distribution, but ideally the delta RTs of the decoys are, to an extent, more or less random and would not be meaningful for optimizing for the targets. Are decoys actually used in the model selection or are they excluded?

Because we only choose the 1000 PSMs with the lowest expectation value, they will only contain target PSMs for most datasets. However, the reviewer is correct that in some cases, such as low input or single-cell data, this PSM subset may contain decoy PSMs, whose delta RT scores should not be minimized. This parameter can be optimized in MSBooster by setting `-numPSMsToCalibrate` to a lower number. Future versions of MSBooster should automatically detect datasets with few PSMs and set this parameter lower. We chose this approach because filtering PSMs, regardless of e-value for targets, has the potential to “overfit” the model to the dataset and violate the equal chance assumption.

We expand upon this in the Method subsection “Heuristic best model search scoring”:

“the 1000/P PSMs with the lowest expectation values (as calculated by MSFragger using the database search hyperscore²⁹) are selected from each pin file, where P is the number of pin files.”

We have corrected the figure caption to say e-value instead of q-value. We have also clarified the final chosen methods:

“The final metrics chosen were “median” for MS2 models and “top consensus” with 10 PSMs for RT models.”

Page 20, Figure 5:

Panel a) is described to display spectral similarity. In our understanding, the y axis would then have to be "unweighted (spectral) entropy similarity", as "spectral entropy" is not a measure for comparing spectra.

The y-axis of Figure 5a is labeled “unweighted spectral entropy”, but we have updated the axis label to span a single line instead of two, in case the text “unweighted” was cut off from the figure.

Panel b): Please clarify whether the x-axis sorted (e.g. "PSM rank").

The caption for Fig 5 has been expanded to describe how the x-axis is sorted: “PSMs are sorted in increasing order of spectral similarity and RT scores for each model (i.e. the lowest delta RT assigned to the 1000 PSMs is plotted at position zero along the x-axis).”

Panel a-b) It is not clear to us why Panel a) directly uses an attribute of the PSM (entropy similarity) whereas for the delta RT axis, a loess smoothing had to be employed.

We clarified in the methods section that we use the loess alignment to mitigate the effects of the experimental setup on the predictions. This is not done for MS2 predictions because they are less affected by experimental variation. The most comparable step to this is the collision energy calibration.

Page 22, Lines 586:

It is appropriately stressed that models scoring best in one metric might not perform best in terms of overall IDs. We would consider it helpful to also discuss that koina models may predict different kinds of ions which may factor into this performance in a way that was not evaluated in the manuscript. E.g. the unispec model predicts many more ions than prosit, and inclusion/exclusion of which affects the spectral similarity calculations.

We agree with the reviewer that the inclusion of non-y/b ions can have a great impact on spectral similarity. Unispec was not used in this manuscript, and MSBooster has options to filter similarity calculation by ion type, but all the models here just used y/b ions. We have added the following to the discussion:

“All models considered in this study only predicted y and b ions. Recent models such as UniSpec can predict other backbone ions (e.g. a-ions and neutral loss peaks), internal fragments, immonium ions, etc. The inclusion of such ions is outside the scope of this study but has the potential to improve peptide identification, especially for non-tryptic peptides.”

Reviewer #2 (Remarks on code availability):

The code is understandable. The online interface is **user-friendly and functional**, and we were able to easily spin up a local server for testing.

The online documentation is easy to read and understand and is generally good. There is some room for improvement in the description of model restrictions, training data, and output. Comments regarding these points are in the main "Comments for author" section.

Reviewer #3 (Remarks to the Author):

The authors present a platform, Koina, designed to enhance accessibility and usability of machine learning or deep learning (ML/DL) models for proteomics research. They aim to lower barriers for laboratories with limited computational resources, facilitating the application of state-of-the-art ML/DL tools in MS-based analyses. While the manuscript addresses a critical need in the proteomics community, certain areas require further elaboration and refinement to meet the standards of Nature Communications.

Major issues:

(1) The development of Koina as a containerized, decentralized, and web-accessible ML platform is a commendable effort to democratize proteomics research. However, the manuscript would benefit from a clearer articulation of Koina's unique contributions compared to existing platforms, such as BioModels.

A side-by-side comparison of functionality, ease of use, and performance metrics would enhance the clarity and impact of the work.

We rewrote the introduction to put more emphasis on the novelty of Koina. We also added a section in the Supplement to compare Koina with BioModelsML and Kipoi.

Also, some case studies are highly recommended to show Koina's unique features. The user case of HLA data analysis actually shows the advantages of the PSM rescoring model, rather than Koina itself.

We politely disagree with the reviewer on this point. The provided model benchmarking over multiple datasets and models shows that different model combinations (from different developers) are required to maximize performance for different datasets and that no single combination always works best. This is expected as specialized models (e.g. for HLA peptides) have a strong tendency to outperform general models. It is further unlikely that a particular model for a particular property will always remain the "best" and that both short- and medium-term adjustments to the processing pipeline will be required to stay up-to-date. Without Koina, a user of FragPipe would have had access only to the DIA-NN RT & MS2 models, which was never the best-performing model combination. Although users theoretically have the option to manually rescore their data with all available models, the effort involved in doing so is

impractical, as most (but not all) models come with their own rescoring pipelines. In contrast with Koina and its integration in FragPipe, the provided Benchmark can be easily repeated by any user for their own data, in a fraction of the time it would have required otherwise.

(2) Some technical implementation requires more detailed explanation. Providing these details would enhance confidence in the platform's high performance.

□ How does Koina ensure scalability and low latency under high user demand?

We clarified in the methods section that we currently use a round-robin load-balancing system that distributes requests equally across all instances: *“Nginx is used to balance the load of the public Koina network between all integrated instances with a round-robin method.”* We also added a section in the outlook on how we aim to improve this load-balancing system to improve performance under high user demand: *“A crucial development to enable a service shared across domains is an improved load balancing mechanism that allows the assignment of models to specific instances. This will also improve performance when multiple users request predictions from different models, since this currently requires loading and unloading of models introducing additional overhead.”*

Ensuring scalability beyond this is not possible, this is a free service that, as of now, is only backed by volunteers who see its value and contribute their available computational resources to be usable by the entire community. Should these computational resources not be sufficient anymore to maintain low latency under high user demand, heavy users would feel encouraged to either self-host a private instance or even contribute their computational resources to the Koina-Network. But apart from deliberate and extensive stress testing, we have not reached a point where the available computational resources are fully used. If Koina is adopted by the community as we expect, we aim to acquire institutional funding and support, e.g. via EBI or ELIXIR/deNBI.

□ What are the benchmarks for server response times, throughput, and model execution speeds under varying conditions?

We performed an in-depth benchmarking of server response times (latency & throughput) depending on model, client-side concurrency, available Koina instances, geographic location, and sequence length (new Supplemental Figures S3 - S14). The results support our previous performance claims.

“These results are consistent with the in-depth throughput and latency benchmark we performed where depending on the used model, available Koina instances and client-side concurrency a throughput of more than 200 000 predictions per second (median >30 000) was achieved. (Supplementary Fig. S3 & S4) with only a subset of the available computational resources. The scaling behavior we observed also shows that with the dedication of additional hardware higher performance can be achieved for most models (Supplementary Fig. S7 - S14).”

□ How does Koina's performance compare when running the same models locally versus via the platform?

Predicting spectra for a full digest of the human proteome (~2 million peptides) takes ~1.5 minutes (AlphaPeptDeep MS2). This is comparable to the performance of running the model on a single Nvidia GTX 1080. Running the same model on CPU takes significantly longer.

(3) I think the heuristic approach for selecting optimal models (e.g., for retention time or MS2 predictions) is practical but limited. The authors are encouraged to explore the integration of Automated Machine Learning (AutoML) into Koina. Currently, the results in this work seem to suggest that there is no objective metric for selecting optimal models. Koina is a platform to streamline model deployment & accessibility, but not to train models. As a result, we believe integrating AutoML into Koina is out of scope. We are sorry for causing this misunderstanding. We adapted the introduction to clarify this: *“The process of a model's development can be structured in the form of an ML lifecycle with different stages: data preparation, model training and optimization, and finally, model deployment. In the scientific context, the model deployment stage is often undervalued, even though it is crucial for generating value for the scientific community.”*

We also adapted the conclusion to clearly mention that training models in the form of refinement learning are not possible with Koina: *“Finally, while fine-tuning models directly within Koina is not possible due to the platform's public and decentralized nature, an integration with an external AutoML framework could feature the direct publication of fine-tuned models on private instances. This would dramatically simplify data analysis with custom ML models allowing a much larger number of scientists to leverage the cutting edge of ML in proteomics.”*

(4) Integration with popular tools like FragPipe and Skyline is a major strength. Koina provides more options for model selection. However, the manuscript should provide more detailed examples or case studies that demonstrate the practical benefits of these integrations in real-world scenarios. Did this flexibility of Koina lead some advantages (such as performance improvement) for end users?

We described the conceptual benefits for users of tools like Oktoberfest, EncyclopeDIA, Skyline and FragPipe in “Koina is easily integrated into third-party software” and “Benchmarking prediction models”. As well as the concrete benefits for users of FragPipe for the number of identified peptides across various HLA datasets in the section “Benchmarking prediction models” (Fig. 4). Additionally, we added new supplemental data showing the benefits for users across various TMT datasets (Supplementary Data S2-3).

Notably, across these datasets, various models perform best. This is expected as specialized models (e.g. for HLA peptides) have a strong tendency to outperform general models. It is further unlikely that a particular model for a particular property will always remain the “best” and that both short- and medium-term adjustments to the processing pipeline will be required to stay up-to-date. This should highlight a major benefit of Koina since only its development has enabled such a comprehensive benchmark. Furthermore, Koina did not only allow us to perform this, but thanks to the integration with MSBooster anybody is able to perform a similar analysis on their own data with minimal effort.

Furthermore, we adapted the discussion of this section to highlight that Koina not only enables this for the currently developed models but for models published on Koina in the future as well: *“These integrations enable users to apply deep learning in application-oriented downstream*

tasks like spectral library generation or PSM rescoring. Notably due to the consistent interface between models, integration with Koina allows seamless interoperability between all models currently available on Koina and models released in the future as well."

Currently, I could not find a clear advantage for bioinformaticians who have programming skills.

We politely disagree with this point. As mentioned by Reviewer 2, which is backed by our experience, installing, fixing, and using a model with an unfamiliar codebase is, unfortunately, a very painful process. We believe that using a model via Koina takes at least an order of magnitude less time and work. We do not believe this to be the case due to inexperience on our part. This time-saving factor is compounded if multiple models are to be tested. Since every model has unique input and output formats. With Koina, the standardization that was in the past required to be done by the user has already been taken care of with the establishment of the common model interface (Fig. 1). This has already been well established as part of this manuscript.

And for biologists who have few programming skills, both the standalone tools and Koina are hard to use for themselves. More detailed tutorials (even including user-guide videos) are needed.

We agree that directly using Koina can be difficult without coding experience. That is why we developed the client libraries for Python and R with examples for both of them being provided on the website. With these clients, anybody with rudimentary coding experience should be able to use Koina. But even with these tools, a potential user needs to be able to execute code. To make directly using Koina more accessible for users with at least rudimentary coding experience, we added a self-guided user tour on the website that helps to familiarize users with how they can directly use Koina.

The so far discussed method all require some level of coding, and we agree that this is likely not the best entry-level for people without any coding experience. For particularly this reason we have already integrated Koina into multiple popular proteomics software packages. We believe that the integration of Koina into MSFragger allows any MS-interested person to benefit from Koina (most of the time likely without their knowledge). Otherwise, we would refer users without any coding experience to the standalone tools whose integration we highlight as part of the manuscript (Oktoberfest, Skyline, EncyclopeDIA and MSBooster). Oktoberfest already has an extensive documentation (<https://oktoberfest.readthedocs.io/en/stable/>) and a number of interactive tutorials. FragPipe, including MSBooster is also extensively documented with user guides for various applications. A specific user guide also goes into detail regarding all available options for Koina. We created new tutorials for Skyline and EncyclopeDIA that show how the Koina integration can be used. Links to all mentioned tutorials are included in the supplementary notes. We also created a Zenodo release of all tutorials to ensure long-term availability.

(5) Another important issue is the authors should make a detailed and clear description about the governance of a community-driven platform (i.e., Koina), including mechanisms for quality control, validation, and versioning of models submitted to Koina.

We developed a contributor guideline specifying our opinions on the points raised above.

<https://github.com/wilhelm-lab/koina/blob/main/CONTRIBUTING.md>

Here is a brief summary of the full contributor's guidelines:

We welcome all contributions, especially encouraged are contributions to improve the documentation models and the release of new models on Koina.

- **Documentation** - *Proper documentation is crucial for users to properly judge a model and its usable range. We have created a documentation questionnaire using the DOME documentation guidelines. Following this guide for writing the documentation of a contributed model is encouraged but not required.*
- **Quality control** - *We don't evaluate models made available on Koina based on their prediction accuracy/performance. This should be discussed in detail with an accompanied preprint/paper. We invite all authors of ML models to contribute their models to Koina and improve the impact of their models by making them more accessible.*
- **Validation** - *We heavily encourage authors of ML models to provide test data for their models. This ensures that predictions are not changing due to unexpected effects of the underlying infrastructure. We have set up a streamlined testing schema that is meant to ensure the long-term reproducibility of models.*
- **Versioning** - *Older versions of models should stay available to enable users to reproduce past results. There are two possible versioning mechanisms supported via Koina. Updates that don't affect the generated predictions, such as performance improvements in pre- and post-processing scripts, don't require a separate version of the model.*
 - *The primary method to version a model is through the numbered folders in the model directory. This ensures that all versions of a model are freely available via Koina. This version mechanism is limited to changes that don't require the config file to be adjusted.*
 - *The second versioning mechanism, in the rare cases where changes to the config file are required, is to create a separate version of the model with an incrementing numbered suffix (<model name>_<version number>)."*

(6) Meanwhile, the decentralized architecture of Koina raises important questions about data security and privacy, particularly for sensitive clinical datasets. The authors should explicitly address:

- How data security is maintained during processing.**
- How did they comply with data protection regulations.**

We adapted the section "Koina is a platform to democratize access to ML models for bottom-up proteomics" to highlight the data security/privacy implications of using Koina with sensitive data: *"The decentralized architecture raises data security questions, particularly for sensitive clinical datasets. To address this, we provide a docker image that facilitates the creation of a self-hosted instance within a secured network, ensuring no external communication. This option allows Koina to be utilized while supporting stringent data security measures."*

We also added a new methods section ("Data security and protection") to clarify what information is collected when the public Koina-Network is used: *"We adhere to data protection*

regulations by collecting only non-identifiable data. Specifically, we gather summary statistics such as source IP addresses, content sizes, and the number of predictions per request, which are necessary for preventing abuse and providing usage statistics for future grant proposals. Importantly, we do not collect the inputs users provide, and these are processed in real time without being stored on non-volatile devices.”

(7) The manuscript’s discussion of future development is limited. A more detailed roadmap, including the addition of AutoML features or other enhancements (e.g., instrument-specific models), would strengthen the paper’s vision for Koina’s growth.

We have extended the outlook in a variety of places to describe where we see potential for future development:

Publication of new state-of-the-art models. While we do not mention this in the manuscript, but we are actively developing additional models that include both specific models for individual instruments, post-translational modifications, and fragmentation methods, as well as generic models that cover multiple instruments, PTMs, and fragmentation methods. All of these will be published on Koina. Other groups we are as yet unaffiliated with also already started to approach us regarding publishing their models on Koina.

Integration with other proteomics software With the model diversity continuously improving, we expect more proteomics data analysis software will choose to integrate with Koina as it provides easy access to a wide variety of ML models that would otherwise need to be manually made compatible with the existing codebase. Which requires significantly more work than working with Koina.

Extension to other domains

“Koina’s approach to making models accessible is not limited to proteomics, it can effortlessly expand to other domains. To exemplify this, we have already integrated two metabolomics models predicting the tandem mass spectra fragmentation of small molecule metabolites. As we consider the optimal structure for this extension, we face the decision of whether to establish independent platforms for each domain or to centralize various models into a singular, adaptable platform. Balancing infrastructure complexity and hardware utilization will be key in this decision.”

Improve performance by developing an intelligent load balancing system.

“A crucial development to enable a service shared across domains is an improved load balancing mechanism that allows the assignment of models to specific instances. This will also improve performance when multiple users request predictions from different models, since this currently requires loading and unloading of models introducing additional overhead.”

Development of an integrated Fine-tuning/AutoML tool. We are already working on this in the form of Dlomix <https://github.com/wilhelm-lab/dlomix>. We are also considering creating a pipeline within FragPipe that would allow for the automated fine-tuning of a model on Koina including the automated release on a private instance of Koina.

“Finally, while fine-tuning models directly within Koina is not possible due to the platforms public and decentralized nature, an integration with an external AutoML framework could feature the direct publication of fine-tuned models on private instances. This would dramatically simplify data analysis with custom ML models allowing a much larger number of scientists to leverage the cutting edge of ML in proteomics.”

Development of an API to communicate the range of supported values for model inputs

“[W]e plan to develop an API to document model limitations in a machine-readable format. Standardizing this in the form of an API improves interoperability between models, allowing downstream data analysis tools to filter their inputs dynamically, giving more flexibility to the user to choose the preferred level of confidence they expect of a model.”

Minor issues:

1. In practical analysis tasks, users may need to fine-tune existing models, and some models (eg. PeptDeep) also support fine-tuning and then prediction tasks. So, does Koina support fine tuning? I didn't see that in the manuscript and Koina's documents.

Koina does not support fine-tuning. This can be performed outside of Koina with the existing codebase of, for example AlphaPeptDeep. The fine-tuned model can then be easily deployed to a self-hosted instance on Koina. Public instances cannot allow this because it opens the service up for arbitrary code execution. But an automatic publication of a fine-tuned model on Koina is possible. We are already working on an AutoML framework called Dlomix (<https://github.com/wilhelm-lab/dlomix>). We are also considering creating a pipeline within FragPipe that would allow for the automated fine-tuning of a model including the automated release on a private instance of Koina. We have adapted the manuscript accordingly: *“Finally, while fine-tuning models directly within Koina is not possible due to the platforms public and decentralized nature, an integration with an external AutoML framework could feature the direct publication of fine-tuned models on private instances. This would dramatically simplify data analysis with custom ML models allowing a much larger number of scientists to leverage the cutting edge of ML in proteomics.”*

2. What is the definition of "top consensus" metric in Fig.5?

This is described in the methods as part of the “Heuristic best model search scoring” section. Here is a copy of this section for your convenience: *“Top consensus: N PSMs with the largest values are selected and sorted. Largest values for the PSM features when testing MS/MS models mean those with the highest spectral similarity; when testing RT models, large values mean the greatest deviations from the RT calibration curve. The values of N tested are 10, 50, and 100. At every position from 1 to N, the model with the best PSM feature value (greatest MS/MS similarity or smallest RT deviation) gets a vote. The model with the most votes was selected.”*

Reviewer #3 (Remarks on code availability):

(1) I were able to install the docker image of Koina on a Linux server. But I suggest considering the compatibility with Windows.

We agree that this would be a valuable feature. Sadly the underlying dependencies that enable Koina only support Linux.

(2) More detailed tutorials (even including user-guide videos) are needed.

Please see our response to the point “And for biologists who have few programming skills, both the standalone tools and Koina are hard to use for themselves. More detailed tutorials (even including user-guide videos) are needed”.

Reviewer #4 (Remarks to the Author):

REVIEWER COMMENTS

Reviewer #1 (Remarks to the Author):

The authors have addressed most of the concerns raised in the previous review, and the manuscript has been significantly improved. However, a few issues remain unresolved and should be clarified before publication:

We thank the reviewer for his positive feedback.

1. While the platform's performance has been demonstrated using synthetic or simulated datasets, its utility must be validated with real-world proteomics data. The authors are suggested to apply their method to experimentally derived protein datasets (e.g., from mass spectrometry or antibody-based assays) to identify bona fide proteins and perform downstream functional analyses (e.g., pathway enrichment, protein-protein interactions). This step is essential to demonstrate practical relevance and robustness.

There appears to be a misunderstanding regarding the data employed in our analysis. We do not use any synthetic or simulated datasets. We utilized 13 real-world mass spectrometry (MS)-based proteomics datasets, which encompass a diverse range of sample types and experimental workflows. These include HLA studies, phosphoproteomics, TMT-labeled samples, body fluids, cell lines, patient tissues, and data from multiple organisms as well as a variety of MS instruments, such as the timsTOF Pro, Q Exactive, Fusion Lumos, Exploris, and Astral.

The utility of machine learning in MS-based data analysis has been well established in the literature (see references 1-7 of the manuscript). This includes significant benefits for downstream functional analyses, as requested by the reviewer. We believe that the analyses we have performed are already sufficient to demonstrate real-world utility. Any additional analyses would not enhance the credibility of this work beyond what has already been demonstrated by us and what is widely accepted in the literature. Nevertheless, we performed an overrepresentation analysis on the phosphoproteins identified with DIA-NN and AlphaPeptDeep predictions in the Arabidopsis dataset (Supplemental Fig. 16). We found most of the overrepresented biological processes were in common between the two, but the data analysis using AlphaPeptDeep enabled via Koina helped identify more phosphoproteins for a more complete view of each process.

2. The manuscript would benefit from a broader perspective on the platform's applicability to cutting-edge fields such as single-cell proteomics and spatial metabolomics.

We have expanded the discussion with our perspective on how the platform supports cutting-edge fields such as single-cell and spatial proteomics.

“Koina's approach is entirely agnostic to the biological or technical source of the dataset, cutting-edge fields such as single-cell or spatial proteomics are natively supported. Furthermore, the platform is not inherently limited to proteomics. It can be effortlessly expanded to other domains. To demonstrate this, we have already integrated two metabolomics models that predict the tandem mass spectra fragmentation of small molecule metabolites.”

Reviewer #2 (Remarks to the Author):

The authors have addressed all concerns. We have one minor comment, explained below. This comment is not blocking publication in our opinion, and the authors can consider if they want to address it when making final edits/revisions.

We thank the reviewer for his positive feedback.

Page 20, Figure 5:

The y-axis of Figure 5a is labeled “unweighted spectral entropy”. We made a comment about this last time but it might have been misinterpreted. We think the label should be "unweighted spectral entropy similarity", the "similarity" part being important because the score is comparing two different spectra. "Entropy" alone would describe the information content of a single spectrum, not the similarity of two spectra. However, it is possible that the use of plain "entropy" as the name of the metric is common in the literature, so we leave it up to the authors to decide how to label the figure.

Thank you for clarifying we agree with the raised points and corrected the axis label.

Reviewer #2 (Remarks on code availability):

The code is understandable. The online interface is user-friendly and functional, and we were able to easily spin up a local server for testing. The online documentation is easy to read and understand. The additions the authors have made to address previous comments are impressive and helpful.

We thank the reviewer for his positive feedback.

Reviewer #3 (Remarks to the Author):

I appreciate that the authors have addressed most of my previous concerns.

We thank the reviewer for his positive feedback.

Here, I only have one minor issue to be considered: What are the theoretical bounds for "top consensus"? Can the authors provide a preferred value?

We updated the corresponding methods section to clarify that the theoretical bounds for this are 0 when a model gets no votes or 1 when a model gets all votes.

Reviewer #3 (Remarks on code availability):

The codes are understandable. The tutorial files are revised and easy to follow.